# PoisoningGuard: Provable Defense against Data Poisoning Attacks to Multi-label Classification

## Abstract

Different from multi-class classification where each testing input only has a single ground truth label, multi-label classification aims to make predictions for testing inputs with multiple ground-truth labels. Multi-label classification has many real-world applications such as disease detection, object recognition, document classification, just to name a few. Recent studies, however, showed that a multi-label classifier is vulnerable to data-poisoning attacks, where an attacker can poison the training dataset of the multi-label classifier such that the classifier makes predictions as the attacker desires. Existing provable defenses are all designed for multi-class classification and they achieve sub-optimal results when applying their robustness guarantees to multi-label classification (as we will demonstrate in this paper). In this work, we propose PoisoningGuard, the *first* provable defense against data-poisoning attacks to multi-label classification. In particular, we generalize two state-of-the-art multi-class certification methods, namely bagging and Deep Partition Aggregation (DPA), to multi-label classification. Our major technical contribution is to jointly consider multiple labels when deriving the provable robustness guarantees. We perform comprehensive evaluations on three datasets. Our experimental results show that our generalized methods significantly outperform bagging and DPA when applying them to multi-label classification. The code will be released.

## 1 Introduction

In *multi-class classification*, an input is assumed to has a single ground-truth label only and thus a multi-class classifier predicts a single label for it. However, in many real-world applications, such as object recognition (Wang et al., 2016), document classification (Partalas et al., 2015), and diseases detection (Ge et al., 2018), an input has multiple ground-truth labels. For instance, an image could contain multiple objects; a patient could be infected with multiple diseases; a document could belong to multiple topics. As a result, multi-class classification is insufficient for those applications. By contrast, *multi-label classification* (Tsoumakas & Katakis, 2007; Trohidis et al., 2008; Read et al., 2009; Wang et al., 2016) assumes each input can have multiple ground-truth labels, and a multi-label classifier predicts multiple labels for it.

Similar to multi-class classification, many recent studies (Ma et al., 2022; Chen et al., 2023; Chan et al., 2023) showed that multi-label classification is also vulnerable to data-poisoning attacks. In particular, given a clean dataset, an attacker could add, delete, and/or modify a certain number of poisoned examples to the clean dataset such that a multi-label classifier makes predictions as the attacker desires. Empirical defenses (Geiping et al., 2021; Peri et al., 2020; Koh et al., 2022; Shokri et al., 2020; Yang et al., 2022; Tran et al., 2018; Liu et al., 2022; Gao et al., 2019; Chou et al., 2020; Chen et al., 2018; Liu et al., 2018; Qiu et al., 2021; Wang et al., 2019; Qiao et al., 2019) cannot provide formal robustness guarantees against data poisoning attacks. Thus, they could be broken by new attacks. For instance, Chen et al. (2023) showed seven state-of-the-art empirical defenses (Guo et al., 2020; Gao et al., 2019; Chou et al., 2020; Chen et al., 2018; Liu et al., 2018; Qiu et al., 2021; Wang et al., 2019; Qiao et al., 2019) are ineffective for poisoning attacks to multi-label classification. Thus, we focus on provable defense in this work.

However, existing state-of-the-art provable defenses (Jia et al., 2020; Levine & Feizi, 2021; Wang et al., 2022) are all designed for multi-class classification. The idea of those defenses as follows. Given a dataset, they first create many sub-datasets that contains a subset of training examples from the given dataset. Then, they train a classifier (called *base classifier*) on each sub-dataset. Finally, they use those base classifier to build an ensemble classifier. In particular, given a testing input, they use each base classifier to predict a label for it and take a majority vote over predicted labels as the final prediction of the ensemble classifier for the given testing input. The predicted label of the ensemble classifier is provably unaffected when the number of added, deleted, and modified samples to a dataset is bounded. As they are designed for multi-class classification, they assume each testing input has a single ground-truth label and they only need to guarantee the predicted label of the ensemble classifier for the testing input does not change when deriving the provable robustness guarantee. By contrast, each input has multiple ground-truth labels in multi-label classification. Thus, the robustness guarantees of those defenses are sub-optimal when applied to multi-label classification as shown in our experimental results.

**Our contribution.** In this work, we propose PoisoningGuard, the *first* provable defense against data poisoning attacks to multi-label classification. In particular, our PoisoningGuard generalizes state-of-the-art certified defenses, namely bagging (Jia et al., 2021) and DPA (Levine & Feizi, 2021), to multi-label classification. Our PoisoningGuard has the following difference with bagging and DPA. First, both base classifier and ensemble classifier in PoisoningGuard can predict multiple labels for a testing input. By contrast, they can only predict a single label in bagging and DPA. Second, given a set of labels, the ensemble classifier of PoisoningGuard can guarantee at least $R$ (called *certified intersection size*) labels in the given set are predicted when the number of added (or deleted or modified) examples to a dataset is no larger than $T$ (called *perturbation size*). By contrast, bagging and DPA can only guarantee their ensemble classifiers provably predict the same label (a single label) for a testing input. Moreover, when the given set only contains a single label, our PoisoningGuard simplifies to bagging and DPA. In other words, they are a special case of our PoisoningGuard.

Our key technical contribution is to derive certified intersection size for a testing input. The major technical challenge is how to jointly consider multiple labels in the derivation. To solve the challenge, we first assume *less than $r$ ground-truth labels are predicted by our PoisoningGuard when a dataset is poisoned (denoted by the predicate $P$)* and then use our assumption to derive a *condition (denoted by the predicate $Q$)*, i.e., $P \rightarrow Q$. By utilizing the *law of contra-position*, i.e., if $P \rightarrow Q$, then $\neg Q \rightarrow \neg P$ ($\neg$ is negation), we know $\neg Q$ is a sufficient condition of $\neg P$, i.e., at least $r$ ground-truth labels are predicted by our PoisoningGuard under poisoning attacks. Given a perturbation size $T$, our certified intersection size is the maximum $r$ such that the sufficient condition is satisfied. In particular, our sufficient condition involves the comparison of two terms, which are very challenging to compute due to 1) the complex training process of base multi-label classifiers (deep neural networks), and 2) those two terms involve multiple labels. In response, we respectively develop new techniques to derive a lower (or upper) bounds for those two terms when our PoisoningGuard is based on bagging and DPA. Specifically, for PoisoningGuard with bagging, we find standard Neyman Pearson Lemma (Neyman & Pearson, 1933) utilized by bagging Jia et al. (2021) cannot be used to jointly consider multiple labels to derive bounds and thus we propose a variant of standard Neyman Pearson Lemma to address the challenge. For PoisoningGuard with DPA, we formulate those two terms as objectives of two constrained optimization problems and then derive lower (or upper) bounds for the objectives.

We perform extensive evaluations on three benchmark datasets. Moreover, we compare our PoisoningGuard with bagging and DPA when applying their robustness guarantees to multi-label classification. Our experimental results show our PoisoningGuard significantly and consistently outperform bagging and DPA in different settings, which demonstrate that jointly considering multiple labels improve the robustness guarantee for multi-label classification. For example, on NUS-WIDE dataset, our PoisoningGuard with bagging ensures that at least 30% of the labels in the ground truth set of testing inputs are correctly predicted on average with perturbation size $T = 100$. Under the same setting, both bagging and DPA can only guarantee 0% of labels are correctly predicted on average under the same setting. Our major contributions are summarized as follows:

- We propose PoisoningGuard, the first provable defense against data poisoning attacks for multi-label classification.

- We derive provable robustness guarantees of PoisoningGuard by jointly considering multiple labels.

- We evaluate our PoisoningGuard on multiple benchmark datasets and compare it with existing state-of-the-art baselines.

## 2 BACKGROUND AND RELATED WORK

**Multi-label classification.** In multi-label classification, a testing input has multiple ground truth labels, and a multi-label classifier predicts a set of labels for it. Multi-label classification has many applications such as medical image diagnosis (Ge et al., 2018), object recognition (Lin et al., 2014), and document classification (Hayes & Weinstein, 1990). Many approaches (Li et al., 2014; Yang et al., 2016; Wang et al., 2016; 2017; Zhu et al., 2017; Nam et al., 2017; Huynh & Elhamifar, 2020; Chen et al., 2019; You et al., 2020; Wu et al., 2020; Ben-Baruch et al., 2020; Dao et al., 2021) have been proposed for multi-label classification. Specifically, a family of studies (Chen et al., 2019; Zhu et al., 2017; Wang et al., 2017; 2016; Li et al., 2014) proposed to design new model architectures. For instance, Wang et al. (Wang et al., 2016) proposed a CNN-RNN framework for multi-label classification, where CNN and RNN parts are used to extract image semantic representations and characterize label relationships, respectively. In general, those methods make complex modifications to model architectures and thus are less general. To address the issue, another family of methods (Wu et al., 2020; Ridnik et al., 2021) proposed to design new loss functions. For instance, Ridnik et al. (2021) proposed an asymmetric loss to handle the positive-negative imbalance (i.e., each class has much more negative samples than positive ones as most images contain a small fraction of possible labels) in multi-label classification. Those methods are agnostic to model architecture and training methods, and are thus they are more general in practice.

**Poisoning attacks to multi-label classification.** Many existing studies (Biggio et al., 2012; Li et al., 2016; Shafahi et al., 2018) showed that multi-class classification is vulnerable to data poisoning attacks (Geiping et al., 2020; Biggio et al., 2012; Li et al., 2016; Shafahi et al., 2018). Similarly, recent studies (Ma et al., 2022; Chen et al., 2023; Chan et al., 2023) showed that multi-label classification is also vulnerable to poisoning attacks. For instance, Chen et al. (Chen et al., 2023) showed that an attacker can poison the training dataset (e.g., mislabel the annotations of certain training images) of a multi-label classifier such that it has attacker desired behaviors, e.g., the multi-label classifier misclassifies images with certain objects. They also evaluate 7 empirical defenses (Guo et al., 2020; Gao et al., 2019; Chou et al., 2020; Chen et al., 2018; Liu et al., 2018; Qiu et al., 2021; Wang et al., 2019; Qiao et al., 2019) generalized from multi-class classification and find that they are ineffective. Thus, we focus on provable defense in this work.

**Existing provable defenses.** All existing provable defenses (Jia et al., 2021; Levine & Feizi, 2021; Steinhardt et al., 2017; Wang et al., 2022; Ma et al., 2019; Rosenfeld et al., 2020; Wang et al., 2020; Zhang et al., 2022) focus on multi-class classification. For instance, bagging (Jia et al., 2021) and DPA (Levine & Feizi, 2021) are two state-of-the-art provable defenses for multi-label classification. However, their robustness guarantees are sub-optimal when applied to multi-label classification as they cannot jointly consider a set of labels. By contrast, our work significantly improves the robustness guarantees for multi-label classification by simultaneously considering multiple labels.

## 3 OUR DESIGN

### 3.1 PROBLEM FORMULATION

**Data poisoning attacks.** Given a clean training dataset $\mathcal{D}_{tr}$, we suppose an attacker can arbitrarily add (or delete or modify) at most $T$ training examples to $\mathcal{D}_{tr}$ to craft a *poisoned training dataset*, and $T$ is the *perturbation size*. For simplicity, we use $\mathcal{S}_p(T, \mathcal{D}_{tr})$ to denote the set of all possible poisoned training datasets when an attacker could add (or delete or modifies) at most $T$ training examples to a clean training dataset $\mathcal{D}_{tr}$. We call an attack *addition attack* (or *deletion attack* or *modification attack*) if an attacker adds (or deletes or modifies) training examples to $\mathcal{D}_{tr}$.

**Certified intersection size.** We suppose $G(\cdot; \mathcal{D}_p)$ is a multi-label classifier trained on a poisoned training dataset $\mathcal{D}_p$. Given a testing input $\mathbf{x}$, $G(\mathbf{x}; \mathcal{D}_p)$ is a set of $k$ labels predicted by the multi-label classifier for $\mathbf{x}$. Suppose $L(\mathbf{x})$ is the set of ground-truth labels of $\mathbf{x}$, given a perturbation size $T$, we define the *certified intersection size* for $\mathbf{x}$ as follows:

$$R(\mathbf{x}; T) = \max r, \ s.t. \ |G(\mathbf{x}; \mathcal{D}_p) \cap L(\mathbf{x})| \geq r, \forall \mathcal{D}_p \in \mathcal{S}_p(T, \mathcal{D}_{tr}). \tag{1}$$

Intuitively, the certified size $R(\mathbf{x}; T)$ is the smallest number of labels in $L(\mathbf{x})$ that are predicted for $\mathbf{x}$ by a multi-label classifier built upon an arbitrary poisoned training dataset in $\mathcal{S}_p(T, \mathcal{D}_{tr})$.

## 3.2 GENERALIZING BAGGING (JIA ET AL., 2021) TO MULTI-LABEL CLASSIFICATION

Below, we first generalize bagging (Jia et al., 2021) to build an ensemble multi-label classifier and then derive its certified intersection size for an arbitrary testing input.

**Ensemble classifier.** Suppose we have a training dataset $\mathcal{D}_{tr}$ that contains $n$ training examples, where each training example consists of an image and a set of ground-truth labels of the image. We can randomly subsample $m$ training examples from $\mathcal{D}_{tr}$ with replacement. For simplicity, we use $\mathcal{X}$ to denote the randomly subsampled training dataset. Given an arbitrary training algorithm $\mathcal{A}$, we can use it to train a base multi-label classifier $g$ on the subsampled dataset, where the hyper-parameter $k_b$ is the number of labeled predicted by $g$ for a testing input. Given a testing input $\mathbf{x}$, we can use $g$ to predict a set of $k_b$ labels for it. For simplicity, we use $g(\mathbf{x}; \mathcal{X})$ to denote the set of $k_b$ predicted labels. Due to the randomness of $\mathcal{X}$, the set $g(\mathbf{x}; \mathcal{X})$ is also random. Given an arbitrary label $l \in \{1, 2, \cdots, C\}$, where $C$ is the total number of classes, we use $p_l$ to denote the probability that the label $l$ is in $g(\mathbf{x}; \mathcal{X})$. Formally, we have $p_l = \Pr(l \in g(\mathbf{x}; \mathcal{X}))$. We call $p_l$ *label probability*. Our ensemble classifier $G$ predicts a set of $k$ labels with the largest label probabilities $p_l$'s for the testing input $\mathbf{x}$. For simplicity, we use $G(\mathbf{x}; \mathcal{D}_{tr})$ to denote the set of $k$ predicted labels. Formally, we have:

$$G(\mathbf{x}; \mathcal{D}_{tr}) = \{l_1, l_2, \cdots, l_k\}$$
$$\text{s.t. } p_{l_i} \geq p_{l_j}, \forall l_i \in \{l_1, l_2, \cdots, l_k\}, \forall l_j \in \{1, 2, \cdots, C\} \setminus \{l_1, l_2, \cdots, l_k\}. \tag{2}$$

Note that $k_b$ and $k$ are the number of predicted labels by a base multi-label classifier $g$ and ensemble multi-label classifier $G$, respectively. Below, we show robustness guarantees of the ensemble classifier.

**Deriving the Certified Intersection Size.** We use $\mathcal{D}_p$ to denote an arbitrary poisoned dataset in $\mathcal{S}_p(T, \mathcal{D}_{tr})$. Moreover, we use $\mathcal{X}$ and $\mathcal{Y}$ to denote two random variables, which represent the randomly subsampled datasets with $m$ training examples from the clean dataset $\mathcal{D}_{tr}$ and the poisoned dataset $\mathcal{D}_p$, respectively. Based on the definition, we have $p_l = \Pr(l \in g(\mathbf{x}; \mathcal{X}))$. We define *poisoned label probability* as $p'_l = \Pr(l \in g(\mathbf{x}; \mathcal{Y}))$, where $l = 1, 2, \cdots, C$. $p'_l$ measures the probability that a label is predicted by a base multi-label classifier trained on a randomly subsampled dataset $\mathcal{Y}$ from the poisoned dataset $\mathcal{D}_p$. Suppose $L(\mathbf{x})$ is a set of $M$ ground truth labels of a testing image $\mathbf{x}$. Our goal is to derive a lower bound on the number of labels (i.e., certified intersection size) in $L(\mathbf{x})$ that are predicted by our ensemble multi-label classifier built upon an arbitrary poisoned dataset $\mathcal{D}_p \in \mathcal{S}_p(T, \mathcal{D}_{tr})$.

The key challenge to directly derive certified perturbation size is how to simultaneously consider multiple labels. To address the challenge, we utilize the *law of contraposition*. Suppose the total number of labels in $L(\mathbf{x})$ that are predicted by our ensemble classifier built upon a poisoned dataset $\mathcal{D}_p$ is smaller than $r$, i.e., the certified intersection size is smaller than $r$, then we know that at least $M - r + 1$ ground truth labels (denoted by $U$) in $L(\mathbf{x})$ are not predicted by our ensemble classifier for $\mathbf{x}$, where $M = |L(\mathbf{x})|$. Similarly, we know at least $k - r + 1$ labels (denoted by $V$) in $\{1, 2, \cdots, C\} \setminus L(\mathbf{x})$ are predicted for $\mathbf{x}$. In other words, we know there exist $U \subseteq L(\mathbf{x})$ and $V \subseteq \{1, 2, \cdots, C\} \setminus L(\mathbf{x})$ such that $\max_{u \in U} p'_u \leq \min_{v \in V} p'_v$. We define the following two predicates: $P$: the certified intersection size is smaller than $r$, and $Q$: $\max_{u \in U} p'_u \leq \min_{v \in V} p'_v$. Our previous derivation shows the following statement is true: if $P$, then $Q$. Based on the law of contraposition, we know if $\neg Q$, then $\neg P$. In other words, if we could show $\max_{u \in U} p'_u > \min_{v \in V} p'_v$ for arbitrary $U$ and $V$, i.e., $\min_U \max_{u \in U} p'_u > \max_V \min_{v \in V} p'_v$, then we know the certified intersection size is no smaller than $r$. Thus, our certified intersection size is the maximum $r$ such that $\min_U \max_{u \in U} p'_u > \max_V \min_{v \in V} p'_v$ is satisfied for an arbitrary $\mathcal{D}_p \in \mathcal{S}_p(T, \mathcal{D}_{tr})$.

It is very challenging to compute $\min_U \max_{u \in U} p'_u$ and $\max_V \min_{v \in V} p'_v$ as our base multi-label classifier is a complex deep neural network. Therefore, we derive a lower bound of $\min_U \max_{u \in U} p'_u$ and an upper bound of $\max_V \min_{v \in V} p'_v$. In particular, suppose $\underline{p_y}$ is a lower bound of probability $p_y$ for for each label $y \in L(\mathbf{x})$ and $\overline{p}_l$ is an upper bound of $p_l$ for each label $l \in \{1, 2, \cdots, C\} \setminus L(\mathbf{x})$. We utilize Neyman Pearson Lemma (Neyman & Pearson, 1933) to derive those bounds. However, the key challenge of standard Neyman Pearson Lemma (Neyman & Pearson, 1933) is that it can only consider each label independently. In other words, it can only be applied when $U$ (or $V$) only

contains a single label. To address the challenge, we develop an extended version of it to jointly consider multiple labels. Due to space reason, we only present high level idea in main text and defer details to Appendix A. Suppose $\Upsilon$ is the joint space between the two random variables $\mathcal{X}$ and $\mathcal{Y}$. We divide $\Upsilon$ into three sub-spaces $A$, $B$, and $C$, i.e., $\Upsilon = A \cup B \cup C$. Based on the definition of $\mathcal{X}$ and $\mathcal{Y}$, we could compute their probability mass functions. Given an arbitrary $U$, we find a sub-space $B' \subseteq B$ such that the probability of $\mathcal{X}$ in that sub-space is $\frac{\sum_{u \in U} p_u^{\#}}{k_b} - \Pr(\mathcal{X} \in A)$, where $p_u^{\#} \triangleq \frac{k_b}{n^m} \lfloor \underline{p_u} \cdot \frac{n^m}{k_b} \rfloor$ (we round $\underline{p_u}$ to make it an integer multiple of $\frac{k_b}{n^m}$ such that $B'$ can be constructed). Then, we derive a lower bound of $\sum_{u \in U} p_u'$ by our extended Neyman Pearson Lemma. Based on the fact that the maximum value in a set is no smaller than the average value, we derive a lower bound of $\max_{u \in U} p_u'$. By considering all possible $U$, we could derive a lower bound of $\min_U \max_{u \in U} p_u'$. Similarly, we derive an upper bound of $\max_V \min_{v \in V} p_v'$. Given those bounds, we can compute the certified perturbation size by letting $\min_U \max_{u \in U} p_u' > \max_V \min_{v \in V} p_v'$ for $\forall \mathcal{D}_p \in \mathcal{S}_p(T, \mathcal{D}_{tr})$. Formally, we have the following theorem:

**Theorem 1** (Certified Intersection Size). *Suppose we have a training dataset $\mathcal{D}_{tr}$. Moreover, we assume we have a base learning algorithm $\mathcal{A}$ that can be used to train a base multi-label classifier on a randomly subsampled $m$ training examples from $\mathcal{D}_{tr}$. Our ensemble classifier $G$ is as defined in Equation 2. Given a testing input $\mathbf{x}$ whose ground truth labels are $L(\mathbf{x}) = \{l_1, l_2, \cdots, l_M\}$. We use $\underline{p_l}$ and $\overline{p}_l$ to denote the label probability lower and upper bound for a label $l = 1, 2, \cdots, C$. Given a perturbation size $T$, we have the following:*

$$|G(\mathbf{x}; \mathcal{D}_p) \cap L(\mathbf{x})| \geq R(\mathbf{x}, T), \forall D_p \in \mathcal{S}_p(T, \mathcal{D}_{tr}), \tag{3}$$

*where $R(\mathbf{x}, T)$ is the solution to the following optimization problem:*

$$R(\mathbf{x}, T) = \arg\max_r r$$

$$s.t. \ \max\left( \max_{t=1}^{M-r+1} \frac{1}{t} \left( \sum_{l=l_r}^{l_{r+t-1}} p_l^{\#} - k_b + k_b \cdot \left(\frac{e}{n}\right)^m \right) \frac{n^m}{(n_p)^m}, \left( p_{l_r}^{\#} - 1 + \left(\frac{e}{n}\right)^m \right) \frac{n^m}{(n_p)^m} \right)$$

$$> \min\left( \min_{t=1}^{k-r+1} \frac{1}{t} \left( \sum_{s=s_{k-r+2-t}}^{s_{k-r+1}} p_s^* \frac{n^m}{(n_p)^m} + k_b \left(1 - \left(\frac{e}{n_p}\right)^m\right) \right), p_{s_{k-r+1}}^* \frac{n^m}{(n_p)^m} + 1 - \left(\frac{e}{n_p}\right)^m \right) \tag{4}$$

*where $p_u^{\#} \triangleq \frac{k_b}{n^m} \lfloor \underline{p_u} \cdot \frac{n^m}{k_b} \rfloor \leq \underline{p_u}$ for $u \in L(\mathbf{x})$ and they satisfy $p_{l_1}^{\#} \geq p_{l_2}^{\#} \geq \cdots \geq p_{l_M}^{\#}$, and $p_v^* \triangleq \frac{k_b}{n^m} \lceil \overline{p}_v \cdot \frac{n^m}{k_b} \rceil \geq \overline{p}_v$ for $v \in \{1, 2, \cdots, C\} \setminus L(\mathbf{x})$. $s_1, s_2, \cdots, s_{k-r+1}$ are the $k-r+1$ labels with the largest $p_v^*$'s in $\{1, 2, \cdots, C\} \setminus L(\mathbf{x})$ and they satisfy $p_{s_1}^* \geq p_{s_2}^* \geq \cdots \geq p_{s_{k-r+1}}^*$. $l_r, l_{r+1}, \cdots, l_M$ are the $M - r + 1$ labels with the smallest $p_u^{\#}$'s in $L(\mathbf{x})$ and they satisfy $p_{l_r}^{\#} \geq p_{l_{r+1}}^{\#} \geq \cdots \geq p_{l_M}^{\#}$. We have $e = n - T$ and $n_p = n$ for modification attack; $e = n$ and $n_p = n + T$ for addition attack; $e = n - T$ and $n_p = n - T$ for deletion attack.*

*Proof.* Please refer to Appendix A. $\square$

We have the following differences with bagging (Jia et al., 2021). First, our base and ensemble classifiers could predict multiple labels while they only only produce a single label in bagging. Second, we jointly consider multiple ground truth labels in our derivation of certified intersection size while bagging can only consider a single label. To jointly consider multiple labels, we utilize new technique such as the law of contraposition and extend standard Neyman Pearson Lemma (Neyman & Pearson, 1933) to consider multiple labels. Our experimental results show our method significantly outperform bagging when extending robustness guarantee in bagging to multi-label classification (the details on the extension is shown in Section 4.1). We note that our certified intersection size reduces to bagging when both $k_b = 1$ and $k = 1$, i.e., bagging is a special case of our method.

**Computing the certified intersection size.** To compute the certified intersection size, we need to estimate probabaility lower or upper bounds of $p_l$, where $l = 1, 2, \cdots, C$. Following previous studies (Cohen et al., 2019; Jia et al., 2021), we utilize Monte-Carlo sampling. We defer the details to Appendix C. Given those probabaility lower or upper bounds, we use binary search to find $R(T, \mathbf{x})$. Algorithm 1 in Appendix shows the complete process.

### 3.3 GENERALIZING DPA (LEVINE & FEIZI, 2021) TO MULTI-LABEL CLASSIFICATION

**Building an ensemble classifier.** Given a training dataset $\mathcal{D}_{tr} = \{\mathbf{x}_i, \mathbf{y}_i\}_{i=1}^n$ with $n$ training examples, we use a hash function $Hash$ to divide it into $N$ sub-datasets, denoted by $\{\mathcal{D}_{tr}^1, \mathcal{D}_{tr}^2, \cdots, \mathcal{D}_{tr}^N\}$. In particular, we view each entry of $\mathbf{x}_i$ or $\mathbf{y}_i$ as a string, concatenate those strings, and feed it to the hash function. For simplicity, we use $Hash(\mathbf{x}_i \oplus \mathbf{y}_i)$ to denote the output of the hash function for the $i$th training example, where $\oplus$ represents concatenation operation. Then, we have $\mathcal{D}_{tr}^j = \{(\mathbf{x}_i, \mathbf{y}_i) \in \mathcal{D}_{tr} | Hash(\mathbf{x}_i \oplus \mathbf{y}_i)\%N = j - 1\}$, where $\%$ is the modulo operator and $j = 1, 2, \cdots, N$. Given each sub-dataset and a training algorithm, we train a base multi-label classifier that outputs $k_b$ labels for a testing input. For simplicity, we use $h_1, h_2, \cdots, h_N$ to denote those base classifiers, where $h_j(\mathbf{x})$ is a set of $k_b$ labels predicted by $h_j$ for $\mathbf{x}$. Given those base multi-label, we build an ensemble multi-label classifier $H$. Given a testing input $\mathbf{x}$, we use $n_l$ to denote the number of base classifiers that predicts the label $l$ for $\mathbf{x}$, i.e., $n_l = \sum_{j=1}^N \mathbb{I}(l \in h_j(\mathbf{x}))$, where $\mathbb{I}$ is the indicator function, $l = 1, 2, \cdots, C$, and $C$ is the total number of classes. We call $n_l$ *label frequency*. Our ensemble classifier $H$ predicts a set of $k$ labels (denoted by $H(\mathbf{x}; \mathcal{D}_{tr})$) with the largest label frequency $n_l$'s for the testing input $\mathbf{x}$. Formally, we have:

$$H(\mathbf{x}; \mathcal{D}_{tr}) = \{l_1, l_2, \cdots, l_k\} \tag{5}$$

$$\text{s.t. } n_{l_i} \geq n_{l_j}, \forall l_i \in \{l_1, l_2, \cdots, l_k\}, \forall l_j \in \{1, 2, \cdots, C\} \setminus \{l_1, l_2, \cdots, l_k\}. \tag{6}$$

Then, we can show that ensemble classifier is provably robust against data poisoning attacks.

**Deriving the certified intersection size.** We use $\mathcal{D}_p$ to denote an arbitrary poisoned dataset in $\mathcal{S}_p(T, \mathcal{D}_{tr})$. Moreover, we use $n_l'$ (called *poisoned label frequency*) to denote the number of base multi-label classifiers that predicts the label $l$ when they are trained on sub-datasets created from a poisoned dataset $\mathcal{D}_p$, where $l = 1, 2, \cdots, C$. Similar to our previous proof, we also utilize the *law of contraposition* to derive the certified intersection size. Suppose $U$ and $V$ are the $M - r + 1$ labels in $L(\mathbf{x})$ and $k - r + 1$ labels in $\{1, 2, \cdots, C\} \setminus L(\mathbf{x})$, respectively. Our goal is to find a maximum $r$ such that $\min_U \max_{u \in U} n_u' > \max_V \min_{v \in V} n_v'$ is satisfied for $\forall \mathcal{D}_p \in \mathcal{S}_p(T, \mathcal{D}_{tr})$ (the reasoning process is similar to our previous proof). For space reasons, we will show high level idea of how we compute $\min_U \max_{u \in U} n_u'$ and $\max_V \min_{v \in V} n_v'$, and defer the details to Appendix B.

For simplicity, we denote $\delta_u = n_u - n_u'$ for $u \in U$. Given a perturbation size $T$, we know an attacker could corrupt at most $\tilde{T}$ sub-datasets, where $\tilde{T} = T$ (or $\tilde{T} = T$ or $\tilde{T} = 2T$) for addition attack (or deletion attack or modification attack). Therefore, at most $\tilde{T}$ base multi-label classifiers change their predictions. As a result, we have the following two observations for an arbitrary $U$: 1) we have $\delta_l \leq \tilde{T}$ since at most $\tilde{T}$ base multi-label classifiers change their predictions for the label $l$, and 2) we have $\sum_{u \in U} \delta_u \leq k_b \cdot \tilde{T}$ since the total number of predicted labels by the $\tilde{T}$ base multi-label classifiers is $k_b \cdot \tilde{T}$. Based on those two observations and the relationship that $\delta_u = n_u - n_u'$, we formulate finding $\max_{u \in U} n_u'$ for an arbitrary $U$ as the following optimization problem:

$$\min_{\{\delta_u | u \in U\}} \max_{u \in U} (n_u - \delta_u), \ s.t. \ \delta_u \leq \tilde{T} \text{ for } \forall u \in U, \text{ and } \sum_{u \in U} \delta_u \leq k_b \cdot \tilde{T}. \tag{7}$$

We note that it is very challenging to solve the optimization problem due to its minimax structure, especially when $T$ is large. To address the challenge, we derive a lower bound for the objective in the optimization problem. We adopt two ways to derive the lower bound. First, we consider each label independently. In this case, we have the following lower bound: $\min_{\{\delta_u | u \in U\}} \max_{u \in U} (n_u - \delta_u) \geq \max_{u \in U} (n_u - \tilde{T})$ since $\forall \delta_u \leq \tilde{T}$. Then, we jointly consider multiple labels. Suppose $U_t$ is a subset of $t$ labels with the smallest label frequencies in $U$, i.e., $U_t \subseteq U$. Based on the fact that the largest value in a set is no smaller than the average value and $\sum_{u \in U} \delta_u \leq k_b \cdot \tilde{T}$, we derive the following lower bound: $\min_{\{\delta_u | u \in U\}} \max_{u \in U} (n_u - \delta_u) \geq \frac{1}{t} \min_{\{\delta_u | u \in U\}} \sum_{u \in U_t} (n_u - \delta_u) \geq \frac{1}{t} (\sum_{u \in U_t} n_u - k_b \cdot \tilde{T})$. This lower bound holds for arbitrary $t = 1, 2, \cdots, M - r + 1$, where $M - r + 1$ is the size of $U$. Thus, we have $\min_{\{\delta_u | u \in U\}} \max_{u \in U} (n_u - \delta_u) \geq \max_{t=1}^{M-r+1} \frac{1}{t} (\sum_{u \in U_t} n_u - k_b \cdot \tilde{T})$. By taking the maximum one of the two bounds, we obtain the final lower bound for $\max_{u \in U} n_u'$. Finally, we find the smallest lower bound of $\max_{u \in U} n_u'$ over different $U$'s as the lower bound of $\min_U \max_{u \in U} n_u'$. Similarly, we could derive an upper bound for $\max_V \min_{v \in V} n_v'$. Given the lower bound of $\min_U \max_{u \in U} n_u'$ and the upper bound of $\max_V \min_{v \in V} n_v'$, we could compute the certified intersection size by finding the maximum $r$ such that the lower bound of $\min_U \max_{u \in U} n_u'$ is larger than the upper bound of $\max_V \min_{v \in V} n_v'$. Formally, we have the following theorem:

**Theorem 2** (Certified Intersection Size). *Suppose we have a training dataset $\mathcal{D}_{tr}$. Moreover, we assume we have a base learning algorithm $\mathcal{A}$ that can be used to train a base multi-label classifier on $N$ sub-datasets $\mathcal{D}_{tr}^1, \mathcal{D}_{tr}^N, \cdots, \mathcal{D}_{tr}^N$ created from $\mathcal{D}_{tr}$. Our ensemble classifier $H$ is as defined in Equation 5. Given a testing input $\mathbf{x}$ whose ground truth labels are $L(\mathbf{x}) = \{y_1, y_2, \cdots, y_M\}$. Without loss of generality, we assume $n_{y_1} \geq n_{y_2} \geq \cdots \geq n_{y_M}$. Given a perturbation budget $T$, we have the following:*

$$|H(\mathbf{x}; \mathcal{D}_p) \cap L(\mathbf{x})| \geq R(\mathbf{x}, T), \forall D_p \in \mathcal{S}_p(T, \mathcal{D}_{tr}), \tag{8}$$

*where $R(\mathbf{x}, T)$ is the solution to the following optimization*

$$R(\mathbf{x}, T) = \arg\max_r r$$

$$s.t. \quad \max_{t=1}^{M-r+1} \frac{1}{t} \left( \sum_{u=l_r}^{l_r+t-1} n_u - \min(k_b \cdot \tilde{T}, t \cdot \tilde{T}) \right) > \min_{t=1}^{k-r+1} \frac{1}{t} \left( \sum_{v=s_{k-r+2-t}}^{s_{k-r+1}} n_v + \min(k_b \cdot \tilde{T}, t \cdot \tilde{T}) \right) \tag{9}$$

*where $\tilde{T} = T$ for addition attack or deletion attack, and $\tilde{T} = 2T$ for modification attack. $s_1, s_2, \cdots, s_{k-r+1}$ are the $k - r + 1$ labels with the largest $n_v$'s in $\{1, 2, \cdots, C\} \setminus L(\mathbf{x})$ and they satisfy $n_{s_1} \geq n_{s_2} \geq \cdots \geq n_{s_{k-r+1}}$. $l_r, l_{r+1}, \cdots, l_M$ are the $M - r + 1$ labels with the smallest $p_u^\#$'s in $L(\mathbf{x})$ and they satisfy $p_{l_r}^\# \geq p_{l_{r+1}}^\# \geq \cdots \geq p_{l_M}^\#$.*

*Proof.* Please refer to Appendix B in supplementary material. $\square$

Our method has the following differences with DPA (Levine & Feizi, 2021). First, both our base and ensemble classifiers could predict multiple labels but they only predict a single label in DPA. Second, we jointly consider multiple labels to derive certified intersection size while DPA only considers a single label. Thus, they achieve sub-optimal results when their robustness guarantee is extended to multi-label classification as shown in our experimental results (the details on the extension is shown in Section 4.1). Second, our technique to derive robustness guarantees is significantly different from DPA. For instance, to joint consider multiple labels, we utilize the law of contraposition and formulate the derivation as an optimization problem as well as derive a lower bound for its objective to find the solution. Our bound reduces to the one in DPA when $k_b = 1$ and $k = 1$, i.e., DPA is a special case of our method.

**Solving the optimization problem in Equation 9.** We solve the optimization problem in Equation 9 efficiently via binary search. Details are shown in Algorithm 1 in Appendix.

## 4 EVALUATION

### 4.1 EXPERIMENTAL SETUP

**Datasets.** We perform evaluation on following benchmark datasets: MS-COCO (Lin et al., 2014), NUS-WIDE (Chua et al., 2009), and VOC-2007 (Everingham et al., 2007). The details of those datasets can be found in Appendix D.

**Base multi-label classifiers.** For all datasets, we use asymmetric loss (ASL) (Ben-Baruch et al., 2020) to train each base multi-label classifier. We use the same hyper-parameter setting as Ben-Baruch et al. (2020). The details could be found in Appendix E. Following previous studies (Jia et al., 2021), we use an encoder pre-trained on ImageNet dataset by MoCo-v2 (Chen et al., 2020) as a feature extractor. In particular, we append a linear layer to the encoder and only optimize the parameters of the least liner layer when training each base multi-label classifier.

**Evaluation metrics.** As evaluation metrics, we utilize *certified top-$k$ precision@$T$*, *certified top-$k$ recall@$T$*, and *certified top-$k$ f1-score@$T$*. These metrics are defined based on a perturbation size $T$, and we describe them as follows: certified top-$k$ precision@$T = R(\mathbf{x}, T)/k$, certified top-$k$ recall@$T = R(\mathbf{x}, T)/|L(\mathbf{x})|$, certified top-$k$ f1-score@$T = 2 \cdot R(\mathbf{x}, T)/(|L(\mathbf{x})| + k)$. As our final result, we report the average values of certified top-$k$ precision@$T$, certified top-$k$ recall@$T$, and certified top-$k$ f1-score@$T$ computed on the testing dataset.

**Compared methods.** We compare our PoisoningGuard-Bagging (PG-Bagging) and PoisoningGuard-DPA (PG-DPA) with state-of-the-art certified defenses against poisoning attacks for

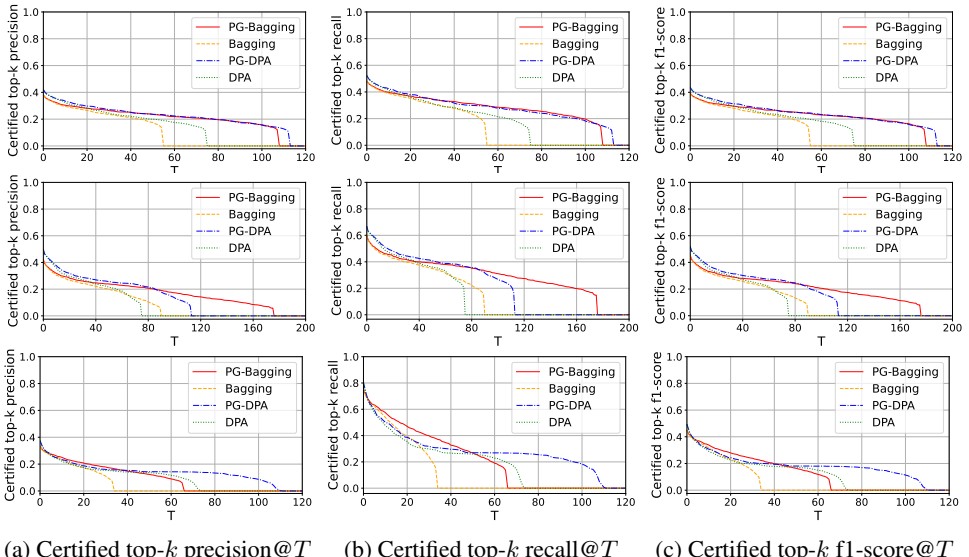

(a) Certified top-$k$ precision@$T$     (b) Certified top-$k$ recall@$T$     (c) Certified top-$k$ f1-score@$T$

**Figure 1: Comparing our methods with bagging and DPA on MS-COCO (first row), NUS-WIDE (second row), and VOC-2007 (third row).**

multi-class classification, including bagging (Jia et al., 2021) and DPA (Levine & Feizi, 2021). We note that all those methods require each base multi-label classifier to predict a single label for each testing input. Thus, we set $k_b = 1$ in our comparison. We use the same model architecture, training algorithm, and hyper-parameters to train base classifiers for different methods to fairly compare them. As bagging and DPA are designed for multi-class classification, they cannot jointly consider multiple labels. In particular, given two labels $l_1$ and $l_2$ where $p_{l_1} > p_{l_2}$ (or $n_{l_1} > n_{l_2}$ for DPA), they can guarantee the poisoned label probability $p'_{l_1}$ (or poisoned label frequency $n'_{l_1}$) is larger than $p'_{l_2}$ (or $n'_{l_2}$) when the number of poisoned examples is bounded. For multi-label classification, we can certify a sample by letting $l_1 = l_r$ and $l_2 = s_{k-r+1}$ (which are defined in Theorem 1). By contrapositive law, proving that the poisoned label probability $p'_{l_r}$ (or poisoned label frequency $n'_{l_r}$) is larger than $p'_{s_{k-r+1}}$ (or $n'_{s_{k-r+1}}$) ensures that the intersection size is at least $r$ when the perturbation size is $T$. Then we apply binary search to find the maximum $r$ such that the inequality $p_{l_r} > p_{s_{k-r+1}}$ (or $n_{l_r} > n_{s_{k-r+1}}$ for DPA) holds. It is worth noting that this is equivalent to solving Equation 4 (or Equation 9 for DPA) with the value of $t$ fixed to 1.

**Parameter setting.** Our primary focus is on modification attacks because they are considered stronger than deletion and addition attacks, as discussed in Jia et al. (2021); Levine & Feizi (2021). We utilize following default hyperparameters. We set $k = 3$ and $k_b = 1$ for all datasets and certification methods. For certifications based on bagging, we set $\alpha = 0.001$ and train 1,000 base classifiers. Additionally, we set $m = 1,000$ for MS-COCO and NUS-WIDE and $m = 100$ for VOC-2007, considering that VOC-2007 has a smaller number of training samples compared to MS-COCO and NUS-WIDE. As for DPA-based certifications, we let $N = 300$ for all datasets. We will study the impact of each hyper-parameter on our method.

## 4.2 EXPERIMENTAL RESULTS

**Comparing PoisoningGuard with existing methods.** Figure 1 shows the comparison results on MS-COCO, NUS-WIDE, and VOC-2007 in default setting. We have two observations. First, we find that PG-Bagging (or PG-DPA) consistently outperforms bagging (or DPA) on all datasets. This is because PoisoningGuard derives robustness guarantees by taking multiple labels into consideration. By contrast, bagging and DPA are designed for multi-class classification and their robustness guarantees are derived by considering a single ground truth label. Second, we can see that PG-Bagging and PG-DPA achieve comparable performances. In particular, each method outperforms the other in certain scenarios. For instance, when the perturbation size $T$ is large, PG-Bagging outperforms PG-DPA on NUS-WIDE but performs worse than PG-DPA on VOC-2007. As for MS-COCO, these two methods show similar performance. We note that PG-Bagging has probabilistic robustness

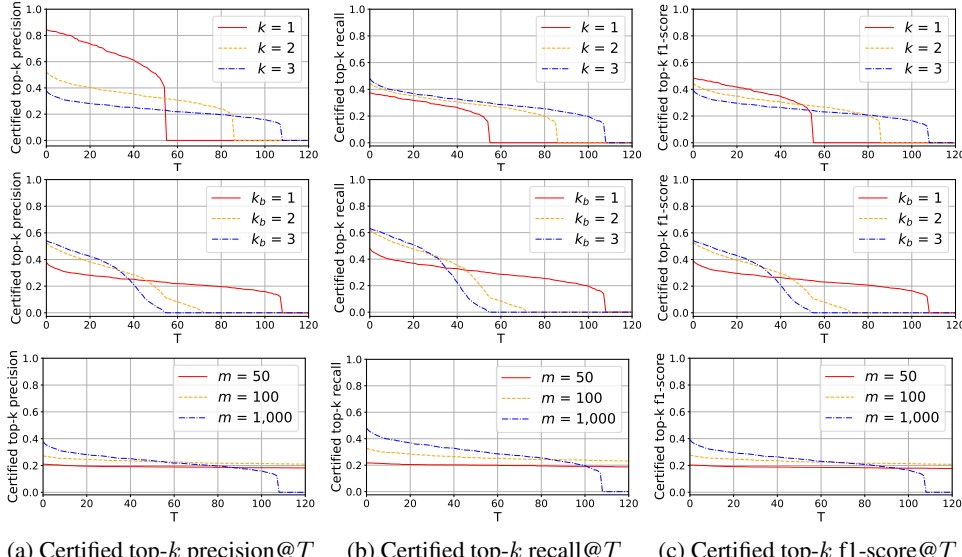

(a) Certified top-$k$ precision@$T$    (b) Certified top-$k$ recall@$T$    (c) Certified top-$k$ f1-score@$T$

**Figure 2: Impact of $k$ (first row), $k_b$ (second row), and $m$ (third row) for PG-Bagging on MS-COCO.**

guarantees (i.e., the certified robustness guarantee is true with a certain probability) but DPA has deterministic robustness guarantees (i.e., the robustness guarantee is true with probability 1).

**Impact of $k$.** The first row of Figure 2 (MS-COCO), Figure 5 (NUS-WIDE; in Appendix) and Figure 7 (VOC-2007; in Appendix) show the impact of $k$ on PG-Bagging. We have the following observations from the results. First, we find that a larger $k$ gives us a smaller certified top-$k$ precision@$T$ without attacks, but the curve drops more slowly as $T$ increases (i.e., a larger $k$ is more robust against poisoned examples as $T$ increases). This is because jointly consider multiple labels is more effective when $k$ is larger. Second, we find that certified top-$k$ recall@$T$ increases as $k$ increases. The reason is that more labels are predicted by our PoisoningGuard as $k$ increases. Note that certified top-$k$ f1-score@$T$ measures a tradeoff between the certified top-$k$ precision@$T$ and certified top-$k$ recall@$T$. We also have those observations for the impact of $k$ on PG-DPA. The results could be found in Figure 4 (MS-COCO), Figure 6 (NUS-WIDE) and Figure 8 (VOC-2007) in Appendix.

**Impact of $k_b$.** The second row of Figure 2 (MS-COCO), Figure 5 (NUS-WIDE; in Appendix) and Figure 7 (VOC-2007; in Appendix) show the impact of $k_b$ on PG-Bagging. We find that PoisoningGuard achieves a larger certified top-$k$ precision@$T$ (or certified top-$k$ recall@$T$ or certified top-$k$ f1-score@$T$) without attacks, but it is less robust as the perturbation size $T$ increases. The reason is that our ensemble classifier could utilize more information from each base multi-label classifier when $k_b$ is large and thus achieve a larger certified top-$k$ precision@$T$ (or certified top-$k$ recall@$T$ or certified top-$k$ f1-score@$T$). However, a larger $k_b$ also leads to additional attack space as each base-multi label classifier predicts more labels, which makes the curves drop more quickly as $T$ increases. We also have those observations for the impact of $k_b$ on PG-DPA. The results could be found in Figure 4 (MS-COCO), Figure 6 (NUS-WIDE) and Figure 8 (VOC-2007) in Appendix.

**Impact of $m$ (or $N$) for PG-Bagging (or PG-DPA).** The third row of Figure 2 (MS-COCO), Figure 5 (NUS-WIDE; in Appendix) and Figure 7 (VOC-2007; in Appendix) show the impact of $m$ on PG-Bagging. We find that $m$ achieves a tradeoff between certified top-$k$ precision@$T$ (or certified top-$k$ recall@$T$ or certified top-$k$ f1-score@$T$) without attacks and robustness, i.e., the curve for a larger $m$ is higher when $T = 0$ but drops more quickly as $T$ increases. The reason is that each base multi-label classifier is more likely to be trained on a poisoned set of subsampled training examples when $m$ is larger. Similar to PG-Bagging, we find that $N$ also achieves a tradeoff between certified top-$k$ precision@$T$ (or certified top-$k$ recall@$T$ or certified top-$k$ f1-score@$T$) without attacks and robustness. The results could be found in Figure 4 (MS-COCO), Figure 6 (NUS-WIDE) and Figure 8 (VOC-2007) in Appendix.

## 5 CONCLUSION

We propose PoisoningGuard, the first certified defense against data poisoning attacks for multi-label classification. Our results show PoisoningGuard significantly improve the robustness guarantees against poisoning attacks for multi-label classification by jointly considering multiple labels.

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
