# A PROOF OF THEOREM 1

We first specify notations and then show our proof. Given a training dataset $\mathcal{D}_{tr}$ and a poisoned training dataset $\mathcal{D}_p$, we respectively use $\mathcal{X}$ and $\mathcal{Y}$ to denote the list of $m$ training examples subsampled from them with replacement. We use $e$ to denote the number of training examples that are in both $\mathcal{D}_{tr}$ and $\mathcal{D}_p$, i.e., $e = |\mathcal{D}_{tr} \cap \mathcal{D}_p|$. Moreover, we use $\Upsilon$ to denote the joint space between $\mathcal{X}$ and $\mathcal{Y}$. We use $\mathcal{E}$ to denote a random variable in the space $\Upsilon$.

We divide the space $\Upsilon$ into the following subspace:

$$A = \{\mathcal{E}|\mathcal{E} \subseteq \mathcal{D}_{tr}, \mathcal{E} \nsubseteq \mathcal{D}_{tr} \cap \mathcal{D}_p\}, B = \{\mathcal{E}|\mathcal{E} \subseteq \mathcal{D}_{tr} \cap \mathcal{D}_p\}, C = \{\mathcal{E}|\mathcal{E} \subseteq \mathcal{D}_p, \mathcal{E} \nsubseteq \mathcal{D}_{tr} \cap \mathcal{D}_p\}, \tag{10}$$

where $\mathcal{E} \subseteq \mathcal{D}_{tr}$ (or $\mathcal{E} \subseteq \mathcal{D}_p$) means every element in $\mathcal{E}$ is in $\mathcal{D}_{tr}$ (or $\mathcal{D}_p$) and $\mathcal{E} \nsubseteq \mathcal{D}_{tr} \cap \mathcal{D}_p$ means there exist at least one element in $\mathcal{E}$ that are not in $\mathcal{D}_{tr} \cap \mathcal{D}_p$. Intuitively, subspace $A$ contains all possible subsampled datasets that can only be obtained from $\mathcal{D}_{tr}$; subspace $B$ contains all possible subsampled datasets that can only be obtained from both $\mathcal{D}_{tr}$ and $\mathcal{D}_p$; subspace $C$ contains all possible subsampled datasets that can only be obtained from $\mathcal{D}_p$.

**Lemma 1.** *Let $\mathcal{X}$, $\mathcal{Y}$ be two random variables whose probability densities are respectively $Pr(\mathcal{X} = \mathcal{E})$ and $Pr(\mathcal{Y} = \mathcal{E})$, where $\mathcal{E} \in \Upsilon$. Let $Z_1, Z_2, \cdots, Z_t : \Upsilon \to \{0,1\}$ be $t$ random or deterministic functions. Let $k'$ be an integer such that:*

$$\sum_{i=1}^{t} Z_i(1|\mathcal{E}) \le k', \forall \mathcal{E} \in \Upsilon, \tag{11}$$

*where $Z_i(1|\mathcal{E})$ denotes the probability that $Z_i(\mathcal{E}) = 1$. Then, we have the following:*

*(1) If $W_1 = \{\mathcal{E} \in \Upsilon : Pr(\mathcal{Y} = \mathcal{E})/Pr(\mathcal{X} = \mathcal{E}) < \mu\}$ and $W_2 = \{\mathcal{E} \in \Upsilon : Pr(\mathcal{Y} = \mathcal{E})/Pr(\mathcal{X} = \mathcal{E}) = \mu\}$ for some $\mu > 0$. Let $S = W_1 \cup W_3$, where $W_3 \subseteq W_2$. If $\frac{\sum_{i=1}^{t} Pr(Z_i(\mathcal{X})=1)}{k'} \ge Pr(\mathcal{X} \in S)$, then $\frac{\sum_{i=1}^{t} Pr(Z_i(\mathcal{Y})=1)}{k'} \ge Pr(\mathcal{Y} \in S)$.*

*(2) If $W_1 = \{\mathcal{E} \in \Upsilon : Pr(\mathcal{Y} = \mathcal{E})/Pr(\mathcal{X} = \mathcal{E}) > \mu\}$ and $W_2 = \{\mathcal{E} \in \Upsilon : Pr(\mathcal{Y} = \mathcal{E})/Pr(\mathcal{X} = \mathcal{E}) = \mu\}$ for some $\mu > 0$. Let $S = W_1 \cup W_3$, where $W_3 \subseteq W_2$. If $\frac{\sum_{i=1}^{t} Pr(Z_i(\mathcal{X})=1)}{k'} \le Pr(\mathcal{X} \in S)$, then $\frac{\sum_{i=1}^{t} Pr(Z_i(\mathcal{Y})=1)}{k'} \le Pr(\mathcal{Y} \in S)$.*

*Proof.* Let's start by proving part (1). For convenience, we denote the complement of $S$ as $S^c$. With this notation, we have the following:

$$\frac{\sum_{i=1}^{t} \Pr(Z_i(\mathcal{Y}) = 1)}{k'} - \Pr(\mathcal{Y} \in S) \tag{12}$$

$$= \int_{\Upsilon} \frac{\sum_{i=1}^{t} Z_i(1|\mathcal{E})}{k'} \cdot \Pr(\mathcal{Y} = \mathcal{E})d\mathcal{E} - \int_{S} \Pr(\mathcal{Y} = \mathcal{E})d\mathcal{E} \tag{13}$$

$$= \int_{S^c} \frac{\sum_{i=1}^{t} Z_i(1|\mathcal{E})}{k'} \cdot \Pr(\mathcal{Y} = \mathcal{E})d\mathcal{E} + \int_{S} \frac{\sum_{i=1}^{t} Z_i(1|\mathcal{E})}{k'} \cdot \Pr(\mathcal{Y} = \mathcal{E})d\mathcal{E} - \int_{S} \Pr(\mathcal{Y} = \mathcal{E})d\mathcal{E} \tag{14}$$

$$= \int_{S^c} \frac{\sum_{i=1}^{t} Z_i(1|\mathcal{E})}{k'} \cdot \Pr(\mathcal{Y} = \mathcal{E})d\mathcal{E} - \int_{S}(1 - \frac{\sum_{i=1}^{t} Z_i(1|\mathcal{E})}{k'}) \cdot \Pr(\mathcal{Y} = \mathcal{E})d\mathcal{E} \tag{15}$$

$$\ge \mu \cdot [\int_{S^c} \frac{\sum_{i=1}^{t} Z_i(1|\mathcal{E})}{k'} \cdot \Pr(\mathcal{X} = \mathcal{E})d\mathcal{E} - \int_{S}(1 - \frac{\sum_{i=1}^{t} Z_i(1|\mathcal{E})}{k'}) \cdot \Pr(\mathcal{X} = \mathcal{E})d\mathcal{E}] \tag{16}$$

$$= \mu \cdot [\int_{S^c} \frac{\sum_{i=1}^{t} Z_i(1|\mathcal{E})}{k'} \cdot \Pr(\mathcal{X} = \mathcal{E})d\mathcal{E} + \int_{S} \frac{\sum_{i=1}^{t} Z_i(1|\mathcal{E})}{k'} \cdot \Pr(\mathcal{X} = \mathcal{E})d\mathcal{E} - \int_{S} \Pr(\mathcal{X} = \mathcal{E})d\mathcal{E}] \tag{17}$$

$$= \mu \cdot [\int_{\Upsilon} \frac{\sum_{i=1}^{t} Z_i(1|\mathcal{E})}{k'} \cdot \Pr(\mathcal{X} = \mathcal{E})d\mathcal{E} - \int_{S} \Pr(\mathcal{X} = \mathcal{E})d\mathcal{E}] \tag{18}$$

$$= \mu \cdot [\frac{\sum_{i=1}^{t} \Pr(Z_i(\mathcal{X}) = 1)}{k'} - \Pr(\mathcal{X} \in S)] \tag{19}$$

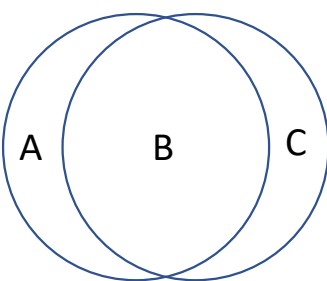

**Figure 3: Illustration Figure.**

$$\geq 0. \tag{20}$$

Equation 16 is derived from 15 due to the fact that $\Pr(\mathcal{Y} = \mathcal{E})/\Pr(\mathcal{X} = \mathcal{E}) \leq \mu, \forall \mathcal{E} \in S$, $\Pr(\mathcal{Y} = \mathcal{E})/\Pr(\mathcal{X} = \mathcal{E}) \geq \mu, \forall \mathcal{E} \in S^c$, and $1 - \frac{\sum_{i=1}^{t} Z_i(1|\mathcal{E})}{k'} \geq 0$. Similarly, we can establish the proof for part (2), but we have omitted the detailed steps for the sake of conciseness. $\square$

For simplicity, we use $n$ and $n_p$ to denote the number of training examples in the training dataset $\mathcal{D}_{tr}$ and poisoned training dataset $\mathcal{D}_p$, i.e., $n = |\mathcal{D}_{tr}|$ and $n_p = |\mathcal{D}_p|$. Then, we have the following probability mass function:

$$\Pr(\mathcal{X} = \mathcal{E}) = \begin{cases} \frac{1}{n^m}, & \text{if } \mathcal{E} \in A \cup B, \\ 0, & \text{otherwise.} \end{cases} \tag{21}$$

$$\Pr(\mathcal{Y} = \mathcal{E}) = \begin{cases} \frac{1}{(n_p)^m}, & \text{if } \mathcal{E} \in B \cup C, \\ 0, & \text{otherwise.} \end{cases} \tag{22}$$

Recall that we have $e = |\mathcal{D}_{tr} \cap \mathcal{D}_p|$, the probability of $\mathcal{X}$ and $\mathcal{Y}$ in those subspace can be computed as follows:

$$\Pr(\mathcal{X} \in A) = 1 - (\frac{e}{n})^m, \Pr(\mathcal{X} \in B) = (\frac{e}{n})^m, \Pr(\mathcal{X} \in C) = 0; \tag{23}$$

$$\Pr(\mathcal{Y} \in A) = 0, \Pr(\mathcal{Y} \in B) = (\frac{e}{n_p})^m, \Pr(\mathcal{Y} \in C) = 1 - (\frac{e}{n_p})^m. \tag{24}$$

We use $M$ to denote the size of $L(\mathbf{x})$ for some test sample $\mathbf{x}$. Suppose this equation $|G(\mathbf{x}; \mathcal{D}_p) \cap L(\mathbf{x})| < r$ is satisfied, then at least $M - r + 1$ ground truth labels in $L(\mathbf{x})$ are not predicted by our ensemble classifier $G$ for $\mathbf{x}$. Similarly, we know at least $k - r + 1$ labels in $\{1, 2, \cdots, C\} \setminus L(\mathbf{x})$ are predicted for $\mathbf{x}$. For simplicity, we respectively use $U$ and $V$ to denote the set of $M - r + 1$ and $k - r + 1$ labels. Since the labels in $V$ are predicted for $\mathbf{x}$ while the labels in $U$ are not predicted, we know there exist $U$ and $V$ such that we have the following:

$$\max_{u \in U} p'_u \leq \min_{v \in V} p'_v \tag{25}$$

Therefore, if

$$\max_{u \in U} p'_u > \min_{v \in V} p'_v \tag{26}$$

, then $|G(\mathbf{x}; \mathcal{D}_p) \cap L(\mathbf{x})| \geq r$. Considering that the selection of $U$ and $V$ is not fixed, it becomes necessary to examine the worst-case scenarios for both $U$ and $V$. Therefore, we derive a lower (or upper) bound on $\min_U \max_{u \in U} p'_u$ (or $\max_V \min_{v \in V} p'_v$). Next, we will derive those bounds.

**Deriving a lower bound on** $\min_U \max_{u \in U} p'_u$**.** For an arbitrary label $u \in U$, we let:

$$p_u^{\#} \triangleq \frac{k_b}{n^m} \lfloor \underline{p_u} \cdot \frac{n^m}{k_b} \rfloor \leq \underline{p_u}, \tag{27}$$

Suppose $U_t$ is a subset of $U$, i.e., $U_t \subseteq U$. Moreover, we define $p_{U_t} = \sum_{u \in U_t} p_u^\#$. We construct a set $S = A + B'$, where $B' \subseteq B$ and $\Pr(\mathcal{X} \in B') = \frac{p_{U_t}}{k_b} - \Pr(\mathcal{X} \in A)$. We can assume $\frac{p_{U_t}}{k_b} > \Pr(\mathcal{X} \in A)$ because otherwise $\mathbf{x}$ cannot be certified by our derived bound. Then, we have the following lower bound on $\max_{u \in U} p_u'$:

$$\max_{u \in U} p_u' \tag{28}$$

$$\geq \max_{U_t \subseteq U} \max_{u \in U_t} p_u' \tag{29}$$

$$\geq \max_{U_t \subseteq U} \frac{1}{t} \sum_{u \in U_t} p_u' \tag{30}$$

$$\geq \max_{U_t \subseteq U} \frac{k_b}{t} \Pr(\mathcal{Y} \in S) \tag{31}$$

$$\geq \max_{U_t \subseteq U} \frac{k_b}{t} \Pr(\mathcal{Y} \in B') \tag{32}$$

$$\geq \max_{U_t \subseteq U} \frac{1}{t} \frac{k_b \cdot \Pr(\mathcal{X} \in B')}{\Pr(\mathcal{X} \in B')} \Pr(\mathcal{Y} \in B') \tag{33}$$

$$\geq \max_{U_t \subseteq U} \frac{1}{t} \frac{p_{U_t} - k_b \cdot Pr(\mathcal{X} \in A)}{\Pr(\mathcal{X} = \mathcal{E} | \mathcal{E} \in B)} \Pr(\mathcal{Y} = \mathcal{E} | \mathcal{E} \in B) \tag{34}$$

$$= \max_{U_t \subseteq U} \frac{1}{t} \cdot (p_{U_t} - k_b \cdot Pr(\mathcal{X} \in A)) \frac{n^m}{(n_p)^m} \tag{35}$$

$$\geq \max_{U_t \subseteq U} \frac{1}{t} (p_{U_t} - k_b + k_b \cdot (\frac{e}{n})^m) \frac{n^m}{(n_p)^m} \tag{36}$$

We have Equation 31 from Equation 30 via Lemma 1, which states that for the set $S$ described above, if $\Pr(\mathcal{X} \in S) \leq \frac{p_{U_t}}{k_b}$, then $\Pr(\mathcal{Y} \in S) \leq \frac{1}{t \cdot k_b} \sum_{u \in U_t} p_u'$. For each $t$, Equation 36 reaches the maximum value when $U_t$ is a set of $t$ labels from $U$ with the largest $p_u^\#$'s. Recall that we have $L(\mathbf{x}) = \{l_1, l_2, \cdots, l_M\}$, where $p_{l_1}^\# \geq p_{l_2}^\# \geq \cdots \geq p_{l_M}^\#$. Thus, $\min_U \max_{u \in U} p_u'$ reaches the minimal value when $U = \{l_r, l_{r+1}, \cdots, l_M\}$. Given $U = \{l_r, l_{r+1}, \cdots, l_M\}$, we have:

$$\max_{u \in U} p_u' \tag{37}$$

$$\geq \max_{U_t \subseteq U} \frac{1}{t} (p_{U_t} - k_b + k_b \cdot (\frac{e}{n})^m) \frac{n^m}{(n_p)^m} \tag{38}$$

$$\geq \max_{t=1}^{M-r+1} \frac{1}{t} (p_{U_t} - k_b + k_b \cdot (\frac{e}{n})^m) \frac{n^m}{(n_p)^m} \tag{39}$$

$$\geq \max_{t=1}^{M-r+1} \frac{1}{t} (\sum_{l=l_r}^{l_{r+t-1}} p_l^\# - k_b + k_b \cdot (\frac{e}{n})^m) \frac{n^m}{(n_p)^m} \tag{40}$$

To consider each element individually, we can construct a set $S = A + B'$, where $B' \subseteq B$ and $\Pr(\mathcal{X} \in B') = p_u^\# - \Pr(\mathcal{X} \in A)$ for any $u \in U$. Then we can again apply Lemma 1 by letting $k' = 1$ and have the following:

$$\min_U \max_{u \in U} p_u' \geq (p_{l_r}^\# - 1 + (\frac{e}{n})^m) \frac{n^m}{(n_p)^m} \tag{41}$$

Putting them together, we have the following:

$$\min_U \max_{u \in U} p_u' \geq \max( \max_{t=1}^{M-r+1} \frac{1}{t} (\sum_{l=l_r}^{l_{r+t-1}} p_l^\# - k_b + k_b \cdot (\frac{e}{n})^m) \frac{n^m}{(n_p)^m}, \tag{42}$$

$$(p_{l_r}^\# - 1 + (\frac{e}{n})^m) \frac{n^m}{(n_p)^m}) \tag{43}$$

**Deriving an upper bound on** $\max_V \min_{v \in V} p'_v$**.**

For an arbitrary label $v \in V$, we have the following:

$$p_v^* \triangleq \frac{k_b}{n^m} \lceil \overline{p}_v \cdot \frac{n^m}{k_b} \rceil \geq \overline{p}_v, \tag{44}$$

Let $p_{V_t} = \sum_{v \in V_t} p_v^*$, we can construct a set $S = C + B'$, where $B' \subseteq B$ and $\Pr(\mathcal{X} \in B') = \frac{p_{V_t}}{k_b} - \Pr(\mathcal{X} \in C)$. Recall that we have $s_1, s_2, \cdots, s_k$ are the $k$ labels with the largest $p_v^*$'s in $\{1, 2, \cdots, C\} \setminus L(\mathbf{x})$. Moreover, we have $p_{s_1}^* \geq p_{s_2}^* \geq \cdots \geq p_{s_k}^*$. Given those two conditions, $\max_V \min_{v \in V} p'_v$ reaches the maximum value when $V = \{s_1, s_2, \cdots, s_{k-r+1}\}$. Suppose $V_t$ is an arbitrary subset of $V$ with $t$ labels. Then, we have the following:

$$\max_V \min_{v \in V} p'_v \tag{45}$$

$$\leq \min_{V_t \subseteq V} \min_{v \in V_t} p'_v \tag{46}$$

$$\leq \min_{V_t \subseteq V} \frac{1}{t} \cdot k_b \cdot \Pr(\mathcal{Y} \in S) \tag{47}$$

$$\leq \min_{V_t \subseteq V} \frac{1}{t} \cdot k_b (\Pr(\mathcal{Y} \in B') + \Pr(\mathcal{Y} \in C)) \tag{48}$$

$$\leq \min_{V_t \subseteq V} \frac{1}{t} \left( \frac{p_{V_t}}{\Pr(\mathcal{X} = \mathcal{E} | \mathcal{E} \in B)} \Pr(\mathcal{Y} = \mathcal{E} | \mathcal{E} \in B) + k_b (1 - (\frac{e}{n_p})^m) \right) \tag{49}$$

$$= \min_{V_t \subseteq V} \frac{1}{t} \left( p_{V_t} \frac{n^m}{(n_p)^m} + k_b (1 - (\frac{e}{n_p})^m) \right) \tag{50}$$

$$\leq \min_{t=1}^{k-r+1} \min_{V_t \subseteq V} \frac{1}{t} \left( p_{V_t} \frac{n^m}{(n_p)^m} + k_b (1 - (\frac{e}{n_p})^m) \right) \tag{51}$$

$$\leq \min_{t=1}^{k-r+1} \frac{1}{t} \left( \sum_{s=s_{k-r+2-t}}^{s_{k-r+1}} p_s^* \frac{n^m}{(n_p)^m} + k_b (1 - (\frac{e}{n_p})^m) \right) \tag{52}$$

By considering each label independently, we have the following:

$$\max_V \min_{v \in V} p'_v \leq p_{s_{k-r+1}}^* \frac{n^m}{(n_p)^m} + 1 - (\frac{e}{n_p})^m \tag{53}$$

Combining them together, we have:

$$\max_V \min_{v \in V} p'_v \leq \min \left( \min_{t=1}^{k-r+1} \frac{1}{t} \left( \sum_{s=s_{k-r+2-t}}^{s_{k-r+1}} p_s^* \frac{n^m}{(n_p)^m} + k_b (1 - (\frac{e}{n_p})^m) \right), \right. \tag{54}$$

$$\left. p_{s_{k-r+1}}^* \frac{n^m}{(n_p)^m} + 1 - (\frac{e}{n_p})^m \right) \tag{55}$$

**Deriving the optimization problem.** By letting $\min_U \max_{u \in U} p'_u > \max_V \min_{v \in V} p'_v$, we have the following:

$$\max \left( \max_{t=1}^{M-r+1} \frac{1}{t} \left( \sum_{l=l_r}^{l_{r+t-1}} p_l^\# - k_b + k_b \cdot (\frac{e}{n})^m \right) \frac{n^m}{(n_p)^m}, (p_{l_r}^\# - 1 + (\frac{e}{n})^m) \frac{n^m}{(n_p)^m} \right) \tag{56}$$

$$> \min \left( \min_{t=1}^{k-r+1} \frac{1}{t} \left( \sum_{s=s_{k-r+2-t}}^{s_{k-r+1}} p_s^* \frac{n^m}{(n_p)^m} + k_b (1 - (\frac{e}{n_p})^m) \right), p_{s_{k-r+1}}^* \frac{n^m}{(n_p)^m} + 1 - (\frac{e}{n_p})^m \right) \tag{57}$$

## B  PROOF OF THEOREM 2

We show that ensemble classifier is provably robust against data poisoning attacks. We use $n_l$ to denote the number of base classifiers that predicts the label $l$ before poisoning and $n'_l$ to denote the

number of base classifiers that predicts the label $l$ after poisoning the training dataset. Suppose the size of $L(\mathbf{x})$ is $M$. When this equation $|H(\mathbf{x}; \mathcal{D}_p) \cap L(\mathbf{x})| < r$ is satisfied, we know that at least $M - r + 1$ ground truth labels in $L(\mathbf{x})$ are not predicted by our ensemble classifier for $\mathbf{x}$. Similarly, we know at least $k - r + 1$ labels in $\{1, 2, \cdots, C\} \setminus L(\mathbf{x})$ are predicted for $\mathbf{x}$. For simplicity, we respectively use $U$ and $V$ to denote the set of $M - r + 1$ and $k - r + 1$ labels. Since the labels in $V$ are predicted for $\mathbf{x}$ while the labels in $U$ are not predicted, we know that we have the following:

$$\max_{u \in U} n'_u < \min_{v \in V} n'_v \vee (\max_{u \in U} n'_u = \min_{v \in V} n'_v \wedge \arg\max_{u \in U} n'_u < \arg\min_{v \in V} n'_v) \tag{58}$$

By contraposition, we know that if

$$\max_{u \in U} n'_u > \min_{v \in V} n'_v \tag{59}$$

for all possible $\{n'_l | l \in \{1, 2, \ldots, C\}\}$, $U$, and $V$, then the test sample is certified. It is essential to derive a lower (or upper) bound on $\min_U \max_{u \in U} n'_u$ (or $\max_V \min_{v \in V} n'_v$). We notice that $U$ includes $M - r + 1$ ground truth labels with smallest $n_u$'s in the worst case, and $V$ includes $k - r + 1$ non-ground truth labels with biggest $n_v$'s in the worst case. As before, we denote the worst case $U$ as $U^* = \{l_r, l_{r+1}, \cdots, l_M\}$, and the worst case $V$ as $V^* = \{s_1, s_2, \cdots, s_{k-r+1}\}$. Then the problem becomes deriving a lower (or upper) bound on $\max_{u \in U^*} n'_u$ (or $\min_{v \in V^*} n'_v$). We denote $|n'_u - n_u|$ as $\delta_u$, then $\max_{u \in U^*} n'_u$ is equivalent as:

$$\min_{\{\delta_u | u \in U\}} \max_{u \in U^*} (n_u - \delta_u) \text{ s.t.} \sum_{u \in U^*} \delta_u \leq k_b \cdot \tilde{T}, \text{ and } \forall u \in U^*, \delta_u \in \mathbb{Z}, \delta_u \leq \tilde{T}. \tag{60}$$

This problem is somewhat complicated due to the min-max structure. As such, it might not have a simple closed-form solution. Here we consider $U_t = \{l_r, l_{r+1}, \cdots, l_{r+t-1}\} \subseteq U^*$, by jointly considering all $u \in U_t$, we can lower bound it by:

$$\min_{\{\delta_u | u \in U\}} \max_{u \in U^*} (n_u - \delta_u) \text{ s.t.} \sum_{u \in U^*} \delta_u \leq k_b \cdot \tilde{T}, \text{ and } \forall u \in U^*, \delta_u \leq \tilde{T}. \tag{61}$$

$$\geq \frac{1}{t} \min_{\{\delta_u | u \in U\}} \sum_{u \in U_t} (n_u - \delta_u) \text{ s.t.} \sum_{u \in U^*} \delta_u \leq k_b \cdot \tilde{T}, \text{ and } \forall u \in U^*, \delta_u \leq \tilde{T}. \tag{62}$$

$$\geq \frac{1}{t} \min_{\{\delta_u | u \in U\}} \left( \sum_{u \in U_t} n_u - \sum_{u \in U_t} \delta_u \right) \text{ s.t.} \sum_{u \in U^*} \delta_u \leq k_b \cdot \tilde{T}, \text{ and } \forall u \in U^*, \delta_u \leq \tilde{T}. \tag{63}$$

$$\geq \frac{1}{t} \left( \sum_{u \in U_t} n_u - \min(k_b \cdot \tilde{T}, t \cdot \tilde{T}) \right) \tag{64}$$

$$\tag{65}$$

If we take the maximum over all $t$, we have:

$$\max_{u \in U^*} n'_u \geq \max_{t=1}^{M-r+1} \frac{1}{t} \left( \sum_{u=l_r}^{l_{r+t-1}} n_u - \min(k_b \cdot \tilde{T}, t \cdot \tilde{T}) \right) \tag{66}$$

We can omit the part that considers each $u \in U^*$ individually since it gives:

$$\max_{u \in U^*} n'_u \tag{67}$$

$$\geq \min_{\{\delta_u | u \in U\}} \max_{u \in U^*} (n_u - \delta_u) \text{ s.t. } \forall u \in U_t, \delta_u \leq \tilde{T}. \tag{68}$$

$$\geq \max_{u \in U^*} (n_u - \tilde{T}) \tag{69}$$

$$\geq n_{l_r} - \tilde{T} \tag{70}$$

, which is equal to $t = 1$ case. Similarly, we can derive:

$$\min_{v \in V^*} n'_v \tag{71}$$

$$\geq \min_{t=1}^{k-r+1} \frac{1}{t} \left( \sum_{v=s_{k-r+2-t}}^{s_{k-r+1}} n_v + \min(k_b \cdot \tilde{T}, t \cdot \tilde{T}) \right) \tag{72}$$

---

**Algorithm 1** Optimize for $r$

---

**Input:** Number of ground truth labels $|L(\mathbf{x})|$, left hand side of the inequality $LHS(\cdot)$, right hand side of the inequality $RHS(\cdot)$.
**Output:** The maximum $r$ s.t. the inequality holds ($LHS(r) > RHS(r)$).
 1: **procedure** OPTIMIZE($|L(\mathbf{x})|$, $LHS(\cdot)$, $RHS(\cdot)$)
 2:  low $\leftarrow$ 0                 ▷ Lowest possible intersection size
 3:  high $\leftarrow |L(x)|$             ▷ Highest possible intersection size
 4:  **while** low < high $-$ 1 **do**
 5:   mid $\leftarrow \lfloor \frac{\text{low}+\text{high}}{2} \rfloor$
 6:   **if** $LHS(\text{mid}) > RHS(\text{mid})$ **then**
 7:    low $\leftarrow$ mid
 8:   **else**
 9:    high $\leftarrow$ mid
10:   **end if**
11:  **end while**
12:  **if** $LHS(\text{high}) > RHS(\text{high})$ **then**
13:   **return** high
14:  **else**
15:   **return** low
16:  **end if**
17: **end procedure**

---

## C Solve Optimization Problem in Equation 4 and Equation 9

**Computing the certified intersection size.** As shown in Equation 4, we need to estimate the probability lower or upper bound for each label to solve the optimization problem to compute the certified size. Following previous studies Jia et al. (2021), we leverage the Monte Carlo algorithm to estimate those probability lower or upper bounds. We randomly sample $N$ subsampled dataset from the training dataset $\mathcal{D}_{tr}$. For simplicity, we use $\mathcal{D}_{tr}^1, \mathcal{D}_{tr}^2, \cdots, \mathcal{D}_{tr}^N$ to denote them. Given those subsampled datasets and a training algorithm $\mathcal{A}$, we can use $\mathcal{A}$ to train a base multi-label classifier on each subsampled dataset. For simplicity, we use $g_1, g_2, \cdots, g_N$ to denote those base classifiers. Given a testing input $\mathbf{x}_{test}$, we can use each base multi-label classifier to predict a label for it. We denote $N_l(\mathbf{x}_{test})$ as the number of base classifiers that predict label $l$ for $\mathbf{x}_{test}$, i.e., $N_l(\mathbf{x}_{test}) = \sum_{i=1}^N \mathbb{I}(f_i(\mathbf{x}_{test}) = l)$, where $l \in \{1, 2, \cdots, C\}$ and $\mathbb{I}$ is the indicator function. Given those $N_l(\mathbf{x}_{test})$'s, we can use Clopper-Pearson method to estimate probability lower or upper bounds. Formally, we can compute them as follows:

$$\underline{p_l} = \text{Beta}(\frac{\alpha}{c}; N_l, N - N_l + 1), \forall l \in L(\mathbf{x}), \tag{73}$$

$$\overline{p}_l = \text{Beta}(1 - \frac{\alpha}{c}; N_l, N - N_l + 1), \forall l \in \{1, 2, \cdots, C\} \setminus L(\mathbf{x}), \tag{74}$$

where $1 - \frac{\alpha}{c}$ is the confidence level and $\text{Beta}(\rho; \varsigma, \vartheta)$ is the $\rho$th quantile of the Beta distribution with shape parameters $\varsigma$ and $\vartheta$.

**Complete algorithm.** Please refer to Algorithm 1 for details.

## D Details of Datasets

Here we provide more details for each dataset.
- **MS-COCO (Lin et al., 2014):** The MS-COCO dataset, also known as Microsoft-COCO (Lin et al., 2014), comprises 82,081 training images, 40,504 validation images, and 40,775 testing images, from a total of 80 different objects. On average, each image within the dataset contains approximately 2.9 objects. It is worth noting that the testing images lack ground truth labels, which means they are not annotated. Therefore, to assess the performance of our method, we follow the previous research (Chen et al., 2019) and evaluate it on the validation dataset.

- **NUS-WIDE (Chua et al., 2009):** The NUS-WIDE dataset, introduced in the paper by (Chua et al., 2009), initially comprises 269,648 images sourced from Flickr. These images are manually annotated across 81 visual concepts, averaging 2.4 visual concepts per image. However, due to some inaccessible image URLs, we utilize the modified version of the dataset released by (Cao et al., 2017). This revised version consists of 134,025 training images and 89,470 testing images, ensuring compatibility and accessibility for our purposes.

- **VOC 2007 (Everingham et al., 2007):** Pascal Visual Object Classes Challenge (VOC 2007) dataset (Everingham et al., 2007) is a widely recognized and extensively utilized benchmark dataset for multi-label classification. The dataset consists of 9,963 images from 20 objects, also known as classes. On average, each image contains 2.5 objects. To align with prior research (Wang et al., 2016), we divided the dataset into 5,011 training images and 4,952 testing images.

## E   BASE CLASSIFIERS

To prevent overfitting due to the limited number of training samples per base classifier, we employ a publicly accessible MoCo-v2 (Chen et al., 2020) encoder that has been pre-trained on ImageNet. This encoder utilizes a ResNet-50 backbone. During the training phase, we supplement the encoder with a linear layer and solely optimizing the linear layer while keeping the pre-trained encoder unchanged.

As for the loss function, we choose Asymmetric loss (ASL) (Ben-Baruch et al., 2020) since this loss addresses the issue of positive-negative label imbalance commonly encountered in multi-label classification. This problem arises when an image has only a few positive labels but numerous negative labels on average.

Let $q_j$ represent the probability that a base multi-label classifier predicts label $j$ $(j = 1, 2, \cdots, c)$ for a given training input. Additionally, let $y_j$ be 1 (or 0) if label $j$ is (or is not) a ground truth label for the training input. The ASL loss [2] is defined as follows: $L_{ASL} = \sum_{j=1}^{c} -y_j L_{j+} - (1 - y_j) L_{j-}$, where $L_{j+} = (1 - q_j)^{\gamma_+} \log(q_j)$ and $L_{j-} = (\max(q_j - \beta, 0))^{\gamma_-} \log(1 - \max(q_j - \beta, 0))$. Here, $\gamma_+, \gamma_-,$ and $\beta$ are hyperparameters.

In accordance with (Ben-Baruch et al., 2020), we set the training hyperparameters as follows: $\gamma_+ = 0$, $\gamma_- = 4$, and $\beta = 0.05$. For training the classifier, we employ the Adam optimizer with a learning rate of 0.01 and a batch size of 32. To carry out our experiments, we utilize the public implementation of ASL[1].

During the testing phase, we report the labels that correspond to the top $k_b$ largest logits.

## F   IMPACT OF $\alpha$ FOR PG-BAGGING

Figure 9 in Appendix show the impact of $\alpha$ on PG-Bagging for the MS-COCO dataset. We find that PG-Bagging achieves slightly better performance as $\alpha$ increases. The reason is the label probability lower or upper bounds are loose when $\alpha$ is large. We also find that our PoisoningGuard is relatively insensitive to $\alpha$, i.e., the influence of $\alpha$ is very small. Our observation is consistent with previous work Cohen et al. (2019); Jia et al. (2021) that utilize Monte-Carlo sampling to compute robustness guarantees. The impact of $\alpha$ on other datasets are shown in the fourth row of Figure 5 and Figure 7.

## G   IMPACT OF ATTACK TYPES FOR POISONINGGUARD

Figure 10 in Appendix show the impact of different attack types on PG-Bagging and PG-DPA for the MS-COCO dataset. We find that our PoisoningGuard achieves better robustness guarantee on addition and deletion attacks than on modification attacks, which means modification attack is the strongest attack type. This finding is consistent with previous works (Jia et al., 2021; Levine & Feizi, 2021).

---

[1]https://github.com/Alibaba-MIIL/ASL

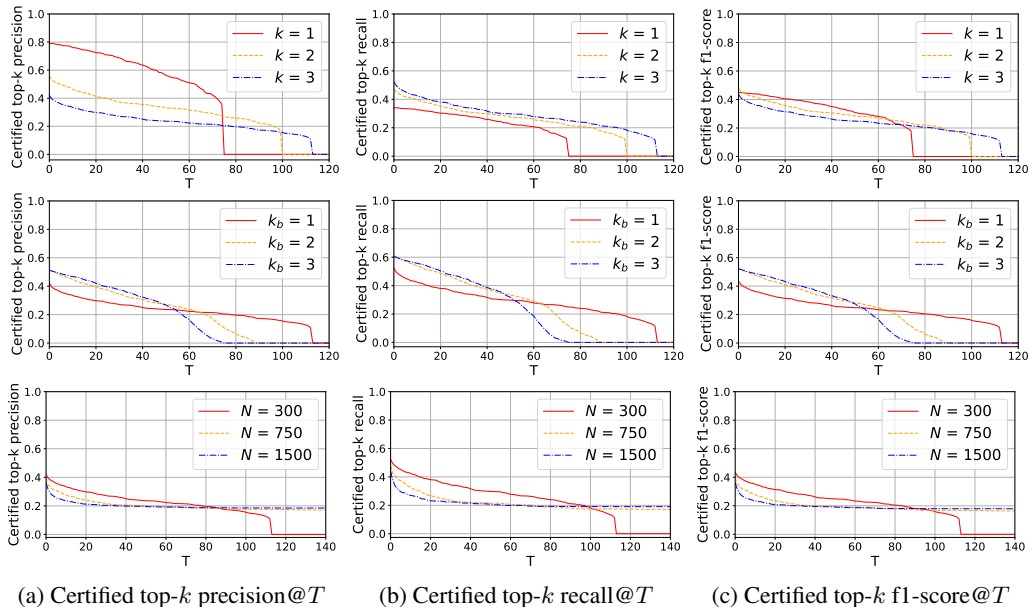

(a) Certified top-$k$ precision@$T$    (b) Certified top-$k$ recall@$T$    (c) Certified top-$k$ f1-score@$T$

**Figure 4: Ablation Study for PG-DPA. Impact of $k$ (first row), $k_b$ (second row), and $N$ (third row). The dataset is MS-COCO.**

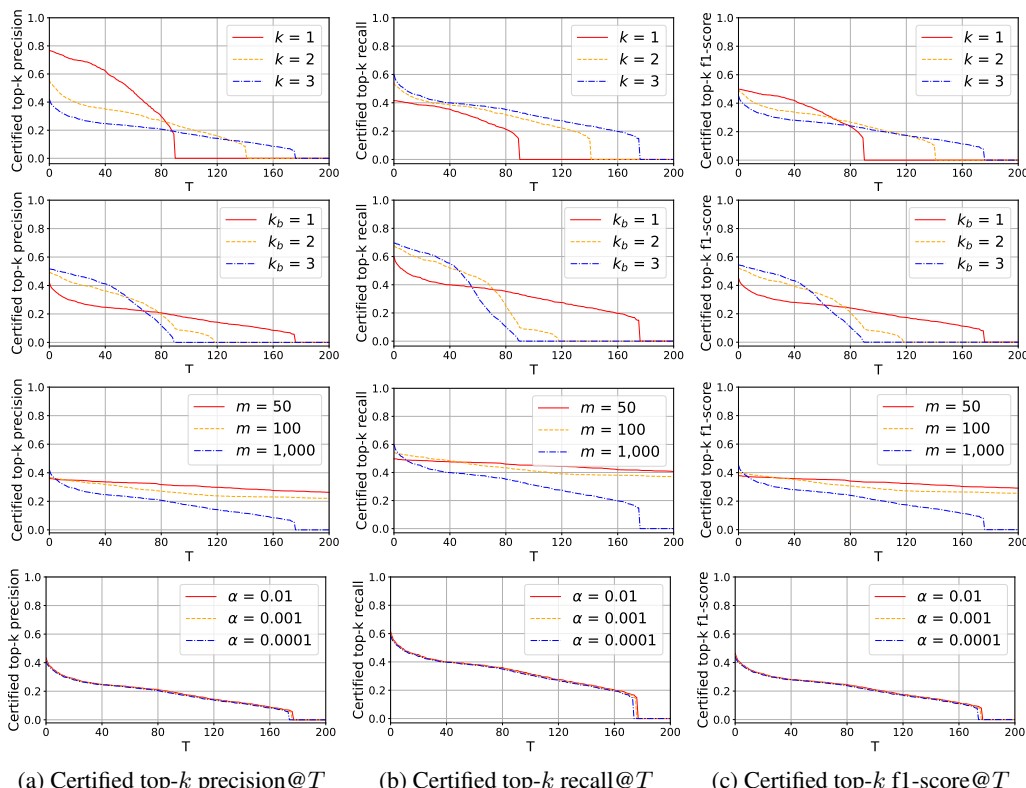

(a) Certified top-$k$ precision@$T$    (b) Certified top-$k$ recall@$T$    (c) Certified top-$k$ f1-score@$T$

**Figure 5: Ablation Study for PG-Bagging. Impact of $k$ (first row), $k_b$ (second row), $\alpha$ (third row), and $m$ (fourth row). The dataset is NUS-WIDE.**

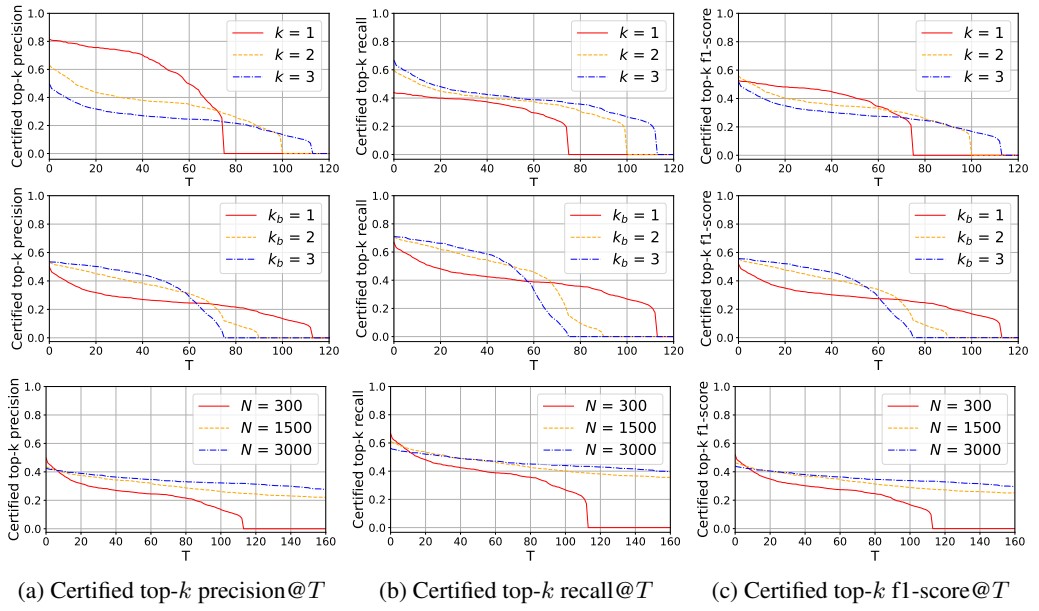

(a) Certified top-$k$ precision@$T$    (b) Certified top-$k$ recall@$T$    (c) Certified top-$k$ f1-score@$T$

Figure 6: Ablation Study for PG-DPA. Impact of $k$ (first row), $k_b$ (second row), and $N$ (third row). The dataset is NUS-WIDE.

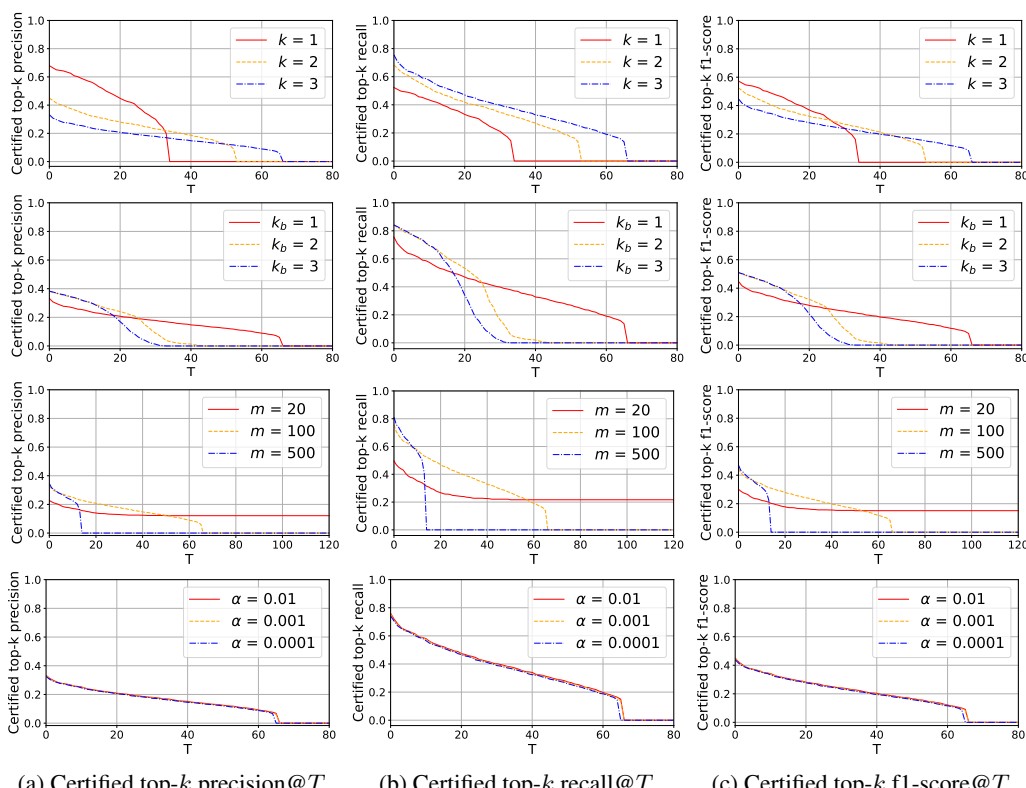

(a) Certified top-$k$ precision@$T$    (b) Certified top-$k$ recall@$T$    (c) Certified top-$k$ f1-score@$T$

Figure 7: Ablation Study for PG-Bagging. Impact of $k$ (first row), $k_b$ (second row), $m$ (third row), and $\alpha$ (fourth row). The dataset is VOC.

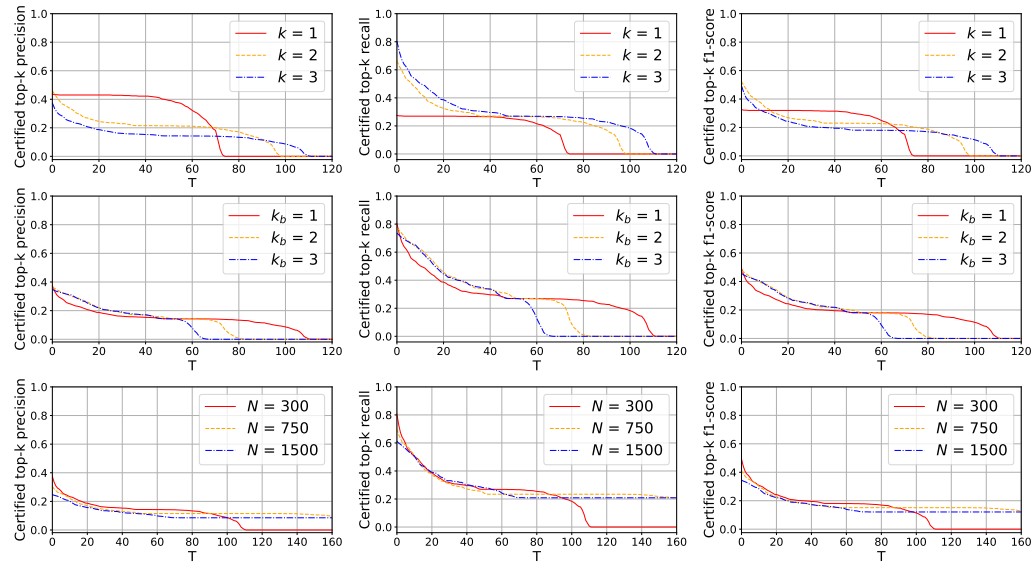

(a) Certified top-$k$ precision@$T$    (b) Certified top-$k$ recall@$T$    (c) Certified top-$k$ f1-score@$T$

**Figure 8: Ablation Study for PG-DPA. Impact of $k$ (first row), $k_b$ (second row), and $N$ (third row). The dataset is VOC-2007.**

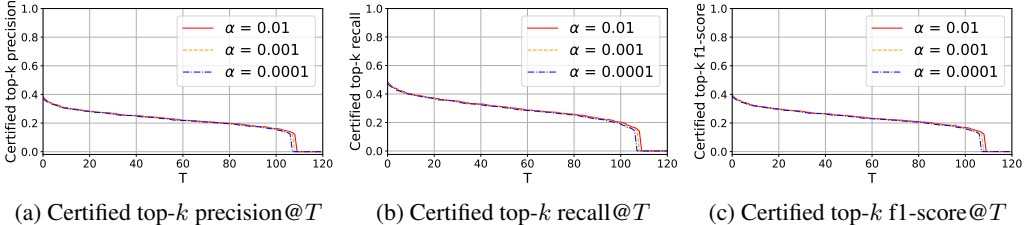

(a) Certified top-$k$ precision@$T$    (b) Certified top-$k$ recall@$T$    (c) Certified top-$k$ f1-score@$T$

**Figure 9: Impact of $\alpha$ for PG-Bagging. The dataset is MS-COCO.**

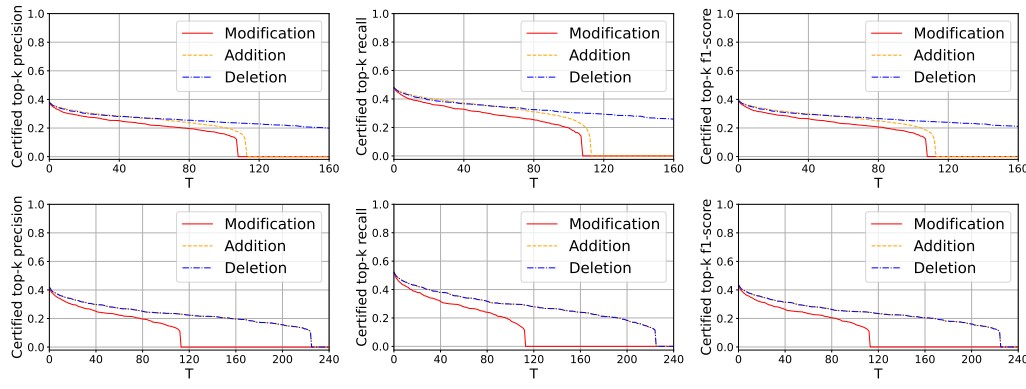

(a) Certified top-$k$ precision@$T$    (b) Certified top-$k$ recall@$T$    (c) Certified top-$k$ f1-score@$T$

**Figure 10: Impact of different attack types on PoisoningGuard. Impact on PG-Bagging (first row) and Impact on PG-DPA (second row). The dataset is MS-COCO.**