# OpenReview forum: "PoisoningGuard: Provable Defense against Data Poisoning Attacks to Multi-label Classification"
_ICLR.cc/2024/Conference — ICLR 2024 Conference Withdrawn Submission_

### Official Review · Reviewer_usg5 · 2023-10-29

**Soundness:** 2 fair
**Presentation:** 2 fair
**Contribution:** 2 fair
**Rating:** 3
**Confidence:** 4

**Summary:**

In this work, two ensemble classification based robust classification methods are proposed by extending the corresponding single-label version of defense methods against data poisoning attacks. The core idea is an extension of Neyman-Perason Lemma from single label to multi-label cases. Theoretical analysis is provided to unveil how many labels are guaranteed to be predicted correctly by applying the proposed robust classification method.

**Strengths:**

Establishing provably robust multi-label classification is an important research problem. Extending Neyman-Pearson lemma to the multi-label case is an interesting angle.

**Weaknesses:**

The weakness in this study is apparent. I will list them as below.

1/ First, both bagging and DPA belong to the ensemble classification-based robust classification mechanism. While it is reasonable to extend them to the multi-label cases, it is unclear why only the ensemble based defense methods are discussed in this study. There are also other ways reaching certifiable or good defense, such as:

**Zhang et al, BagFlip: A Certified Defense Against Data Poisoning, NeurIPS 2022 **

**Liu et al, Friendly Noise against Adversarial Noise: A Powerful Defense against Data Poisoning Attacks, https://arxiv.org/abs/2208.10224**

Even for DPA, there is a recent improved version with certifiable robustness guarantee:

**Wang et al, Improved Certified Defenses against Data Poisoning with (Deterministic) Finite Aggregation, ICML 2022**

It would be useful to at least explain why other data poisoning methods are not discussed or compared with. The reason behind this request is: For real-world practices, the primary expectation is to expect a good defense performance (while I agree setting up certifiable robustness is interesting). This is also the core motivation of proposing any novel dense methods against data poisoning. In this sense, in this study, it is unclear how the proposed method's defense strength look like compared to the state-of-the-art defense methods.

2/ Data poisoning includes clean / dirty-label poisoning and label flipping. It looks the proposed method only considers dirty-label poisoning scenarios. How would the proposed method perform under other data poisoning scenarios ?

3/ Which attack algorithm are used in the experimental study? If we vary the poisoning budget (the number of poisoned training samples and the modification magnitude over the poisoned training samples), how would the proposed defense method perform?

4/ I agree with the law of contra-position. However, the negation of Q is a sufficient condition of the negation of P, not a necessary one.  Therefore, maximizing $r$ following the constraints in Eq.4 and 9 may pose too strict restriction, which may not help reach optimal defense in practices.

All in all, I appreciate the theoretical efforts that this study makes to harden multi-label classifiers. Nevertheless, the current form of the study is only a simple extension of two ensemble classification-based defense methods. It lacks deeper discussion for organising an effective data poisoning defense.

**Questions:**

Please check the raised concern / requests in the comments.

---

### Official Review · Reviewer_PaSr · 2023-10-31

**Soundness:** 2 fair
**Presentation:** 3 good
**Contribution:** 2 fair
**Rating:** 5
**Confidence:** 3

**Summary:**

The paper introduces a new, provable defense method against data poisoning in multi-label tasks. The approach is based on bagging and Deep Partition Aggregation (DPA), proposing the use of ensemble models capable of outputting multiple labels as a defense against data poisoning.

**Strengths:**

- The manuscript is well-organized and easy to follow.
- The authors present the first provable defense method that is specifically designed for multi-label data poisoning scenarios.
- The paper proves that, under the constraint that the number of poisoned data samples is below a threshold 'T,' the method can guarantee 'R' correct predictions.

**Weaknesses:**

- The work builds upon existing defense methods to adapt them for multi-label models, which lessens the overall contribution and novelty of the paper.
- The derivation of the "certified intersection size" is not clearly elucidated, leading to some confusion.
- The paper modifies both bagging and DPA, resulting in a dispersed focus and unclear main points.
- The volume of experiments is insufficient. More results are needed to substantiate the paper's claims.
- The experimental setup lacks clarity.

**Questions:**

- What is the computational cost associated with implementing this defense method?
- Does using an ensemble model impact the original clean accuracy of the base models?
- What data poisoning attacks are employed in the experiments?

---

### Official Review · Reviewer_hayp · 2023-11-01

**Soundness:** 2 fair
**Presentation:** 3 good
**Contribution:** 2 fair
**Rating:** 5
**Confidence:** 5

**Summary:**

This paper adapt some existing provable defenses against data poisoning attacks (i.e. bagging and DPA) that are previously developed in single-label classifications to the multi-label classification settings.

It generalizes bagging and DPA by not only outputting the top-$k$ classes for the ensemble but also allowing each base classifiers of the ensemble to contribute $k_b$ votes.

They evaluate the certificates on MS-COCO, NUS-WIDE and VOC-2007.

**Strengths:**

Overall the quality of this paper is good.
1. Developing provable defenses against data poisoning attacks is an important topic.
2. The multi-label classification angle is interesting.
3. While I did not check the proofs line by line, I am familiar with these provable defenses that I think these results are reasonable.

**Weaknesses:**

1.My major concern is in fact **a very specific mismatch** between the proposed defense and the setting of multi-label classifications.
Specifically, the proposed defense **always predict k labels for a pre-defined constant k**, which is unnatural in multi-label classification. In typical multi-label classification, one does not really assume every sample must have exactly the same number of labels. For this reason, I am concerned **whether this is a reasonably good start for provable defenses in multi-label classification**.

In my opinion, the current formulation is more related to certifying top-k accuracy rather than multi-label classification. However, it is relatively trivial (I will still consider this as part of this work's contribution though) to generalize bagging/DPA for b to top-k accuracy since it is somewhat straightforward (at least for DPA) to allow each base classifiers to contribute multiple votes.

2.There are some closely related but missing references [1-6 below]. [1-5] are closely related to provable defenses against poisoning attacks and [6] is about provable defenses for top-k predictions but for test-time attacks. Notably, the certificates of [4] essentially offer a solution for certifying top-2 predictions. Note that some of these references are in fact fairly new, thus while I suggest authors to discuss them, it is understandable that they are missing from the submitted version.


[1] Chen R, Li Z, Li J, Yan J, Wu C. On Collective Robustness of Bagging Against Data Poisoning.

[2] Wang W, Levine A, Feizi S. Lethal Dose Conjecture on Data Poisoning.

[3] Wang W, Feizi S. Temporal Robustness against Data Poisoning.

[4] Rezaei K, Banihashem K, Chegini A, Feizi S. Run-Off Election: Improved Provable Defense against Data Poisoning Attacks.

[5] Wang, W. and Feizi, S. On Practical Aspects of Aggregation Defenses against Data Poisoning Attacks.

[6] Jia, J., Wang, B., Cao, X., Liu, H. and Gong, N.Z. Almost tight l0-norm certified robustness of top-k predictions against adversarial perturbations.

**Questions:**

Please refer to Weakness.

1. The major concern is about Weakness 1.
Please clarify if you think there is anything missing.
If you also find this to be reasonable, a direct fix from me will be to use either a fixed threshold or an adaptive one to decide whether each label is predicted or not. For example, instead of predicting a constant of k labels per sample, the ensemble can predict all classes with more than t votes for some constant t.
This can actually simplify the computation of certificates and potentially make it much more suitable for the setting of multi-label classifications. However, this leads to critical changes to the proposed approach so revising&resubmiting to future venues may also be a good option.

---

### Official Review · Reviewer_oRDf · 2023-11-03

**Soundness:** 2 fair
**Presentation:** 2 fair
**Contribution:** 3 good
**Rating:** 5
**Confidence:** 3

**Summary:**

This paper extends the existing provable defenses (Bagging and DPA) against data poisoning attacks to the setting of multi-label classification. Based on the notion of certified intersection size $R(x; T)$, the proposed defense provides a certificate that the predicted labels overlap with the ground truth with at least $R(x; T)$ labels, for any poisoned dataset with T poisons. Empirical results show that the proposed PoisoningGuard outperforms baseline provable defenses that are designed for multi-class classification.

**Strengths:**

Developing effective provable defenses against data poisoning attacks in the context of multi-label classification is important. The paper did a good job of presenting the problem formulation, introducing the concept of certified intersection size, and explaining the technical aspects of their certification methods. The paper is also technically sound and the high-level proof roadmap with respect to the main theorems is explained clearly. I like the idea of using the law of contraposition to derive the certification bound.

**Weaknesses:**

Although presenting the technical details for theoretical results is very important, the main theoretical sections (Section 3.2 and Section 3.3) are loaded with too many mathematical notations. I have to read back and forth to pinpoint the meaning of those mathematical notations. Also, the statements of Theorem 1 and Theorem 2 are long and hard to parse to some extent. I suggest presenting all the necessary mathematical notations at the beginning of those sections, using minimal notations when explaining the high-level proof techniques, and simplifying the theoretical statements (or presenting a shorter version of the theorems in the main paper while deferring the more extended version to the appendix).

My major concern about the paper is the empirical comparisons. From my perspective, the experimental sections are weak, which should be expanded to be more comprehensive. The comparisons between the proposed methods with baseline methods (Bagging and DPA) are only conducted for the fixed setting of $k=3$ and $k_b=1$ (not to say other hyperparameters, such as m and N, will also affect the certification performance). It needs to be clarified what the comparison is when those parameters are varied. To convincingly support your claim, I expect more comprehensive experiments to demonstrate the comparisons of your provable defenses with others. Since the baseline methods are designed for multi-class classification which is a different setting, I would like to know how you select the hyperparameters for their method for fair comparisons.

Typos: page 1: assumed to has -> assumed to have; page 5: only only -> only

**Questions:**

Moreover, I have the following questions related to the proposed certification method:

1. To obtain the certified intersection size (e.g., Theorem 1 w.r.t. bagging), you need to solve the constrained optimization theoretical and resort to Monte-Carlo samples to provide empirical estimates. How can you guarantee the soundness of the derived certificate if it is based on the empirical estimates of the underlying probabilities? Is the certification a probabilistic guarantee like randomized smoothing (Cohen et al., 2019)?

2. I would like to know the computational complexity of your certification methods, with comparisons to the corresponding provable defenses (Bagging or DPA). What is the computational overhead of your proposed certification techniques?

3. By examining the empirical results in Figure 1, all the certified defenses against data poisoning attacks sacrifice a large amount of clean performance (T=0). Under most settings, the precision/recall curve is fairly smooth (in other words, decreasing slowly). Does this result mean that all of these provable defenses are overly smooth? Given the low performance when there are no poisons, practitioners may not be interested in deploying them in practice). Is there a way to further improve the clean performance of these provable defenses?

4. Finally, I wonder how to set the best hyperparameters for your certified methods to maximize the performance for a given multi-label classification task. Can you provide some insights?